# SURFACEBENCH: A Geometry-Aware Benchmark for Symbolic Surface Discovery

**Sanchit Kabra** *Department of Computer Science, Virginia Tech*      *sanchit23@vt.edu*

**Shobhnik Kriplani** *Department of Computer Science, Virginia Tech*

**Parshin Shojaee** *Department of Computer Science, Virginia Tech*

**Chandan K. Reddy** *Department of Computer Science, Virginia Tech*

**Reviewed on OpenReview:** *https://openreview.net/forum?id=sHLTzkczSi*

## Abstract

Equation discovery from data is a central challenge in machine learning for science, which requires the recovery of concise symbolic expressions that govern complex physical and geometric phenomena. Recent large language model (LLM) approaches have shown promise in symbolic regression, yet existing benchmarks predominantly evaluate low-dimensional scalar functions and rely on string-level or regression-based metrics that fail to capture structural and geometric equivalence. We introduce SURFACEBENCH, the first geometry-aware benchmark for symbolic discovery of three-dimensional surfaces. Unlike scalar curve-fitting tasks, SURFACEBENCH targets surface-level reasoning, where multi-variable coupling, coordinate transformations, and geometric structure must be inferred directly from data. The benchmark comprises *183* analytically constructed, science-inspired surface equations across *15* categories and three representation paradigms: explicit, implicit, and parametric forms. Each task includes variable semantics and synthetically sampled 3D data, and is designed to stress symbolic composition, structural ambiguity, and representational non-uniqueness while mitigating memorization. To evaluate discovery quality, SURFACEBENCH incorporates symbolic equivalence checks with geometric metrics of the object-space (Chamfer and Hausdorff distances) and regression-based error measures, allowing the evaluation of functional fidelity beyond algebraic syntax. Empirical evaluation across evolutionary, neural, and LLM-driven frameworks reveals that no current method achieves consistent performance across representation types, with LLM-based approaches exhibiting strong structural priors but limited robustness in parameter calibration and multi-equation reasoning. SURFACEBENCH provides a challenging and diagnostic testbed that bridges symbolic reasoning and geometric reconstruction, enabling principled benchmarking of compositional generalization and structure-aware scientific induction in high-dimensional equation discovery. The code and data are available at this link: github.com/deep-symbolic-mathematics/surfacebench.

## 1 Introduction

Symbolic regression, or equation discovery, lies at the core of machine learning for scientific discovery (Reddy & Shojaee, 2025). Its objective is to recover interpretable mathematical expressions that explain observed data and reveal the governing structure of physical, biological, and engineered systems. By transforming raw observations into symbolic form, symbolic regression enables the identification of functional relationships, invariances, and compositional structure that underlie complex phenomena.

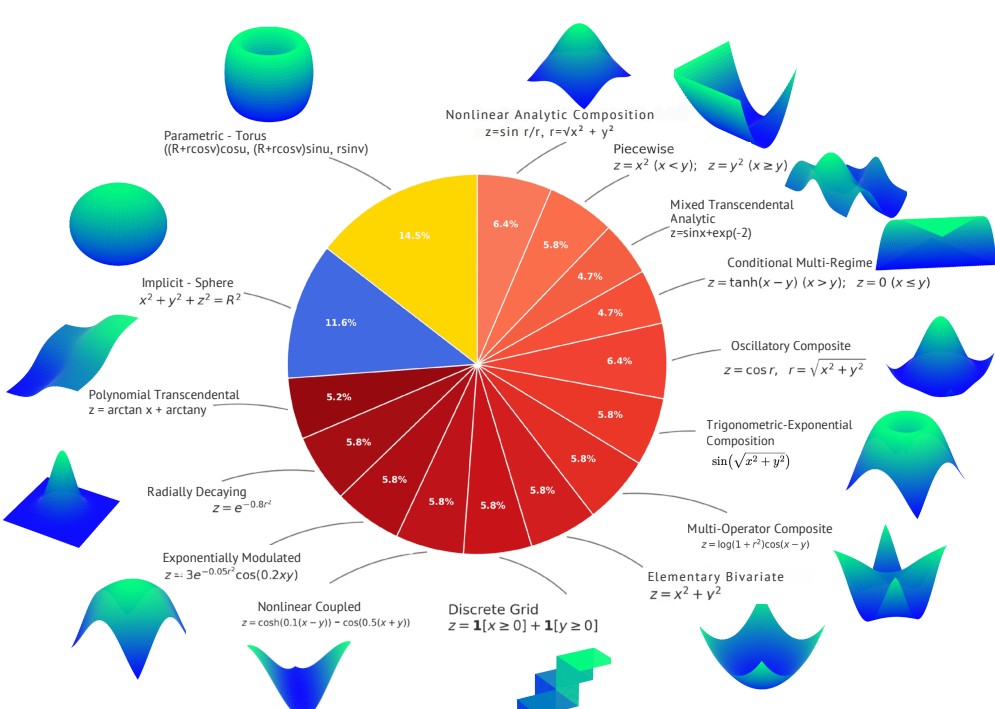

Figure 1: SURFACEBENCH: A benchmark suite for symbolic regression featuring 183 surface equations spanning 15 structurally defined categories. The benchmark covers explicit (red), implicit (blue), and parametric (yellow) representations, illustrating diverse symbolic and geometric challenges.

Traditional approaches relied heavily on domain experts to manually specify relevant variables, transformations, and functional templates (Brunton et al., 2016). While scientifically grounded, such workflows are labor-intensive and difficult to scale. Automated alternatives based on genetic programming, evolutionary search, and neural sequence modeling (Cranmer, 2023; Biggio et al., 2021; Shojaee et al., 2023; Petersen et al., 2019; Landajuela et al., 2022; Kamienny et al., 2022) alleviate manual design but face substantial combinatorial complexity and often require extensive search to navigate vast hypothesis spaces. More recently, large language models (LLMs) have been introduced as structural priors for symbolic discovery. Frameworks such as LLM-SR (Shojaee et al., 2025a), LaSR (Grayeli et al., 2024), SGA (Ma et al., 2024), and OpenEvolve (Sharma, 2025) leverage pretrained knowledge to bias search toward plausible functional families and improve efficiency. However, two fundamental limitations remain. First, LLM-based approaches are susceptible to memorizing canonical expressions rather than reasoning from data. Second, autoregressive generation struggles to tightly couple discrete structural exploration with continuous parameter calibration, limiting robustness in complex scientific regimes.

Despite rapid methodological advances, benchmarking symbolic regression remains fundamentally limited. Existing benchmarks are either synthetic (Shojaee et al., 2025b) or curated from canonical textbook equations (Udrescu & Tegmark, 2020), making them vulnerable to memorization and narrow in structural diversity. Moreover, current benchmarks predominantly evaluate scalar mappings of the form $y = f(x)$. Such tasks fail to reflect the *multivariate coupling, geometric structure, and representational diversity characteristic of real scientific equations.* More critically, commonly used evaluation metrics, such as string matching and normalized mean squared error (NMSE), are fundamentally inadequate for higher-dimensional symbolic objects due to *symbolic non-uniqueness.* For example, a sphere can be written implicitly as $x^2 + y^2 + z^2 = R^2$, explicitly as $z = \pm\sqrt{R^2 - x^2 - y^2}$, or parametrically via trigonometric coordinates; all describe identical geometry yet differ algebraically. Metrics operating solely in symbolic or pointwise regression space therefore fail to capture functional equivalence in geometric settings. As a consequence, strong performance on existing benchmarks does not necessarily imply robust symbolic reasoning or structural generalization.

In contrast, *surface equations fundamentally alter the nature of equation discovery.* Surfaces admit multiple representation paradigms (explicit, implicit, and parametric) each *introducing distinct symbolic, geometric,*

*and topological challenges.* Recovering a surface equation from sampled data $(x, y, z)$ requires *reasoning over coupled outputs, latent coordinate systems, invariances, and structural ambiguity.* Unlike scalar curve fitting, surface-level discovery demands simultaneous reasoning about algebraic structure and spatial consistency, as well as evaluation in object space rather than purely symbolic space.

To address these gaps, we introduce SURFACEBENCH (Figure 1), the first systematic benchmark for symbolic surface discovery. SURFACEBENCH comprises 183 surface equations spanning 15 structurally defined categories and three canonical representation forms: explicit, implicit, and parametric. The benchmark reframes equation discovery from scalar regression to geometry-aware multi-output reasoning. Each task provides analytically constructed surfaces inspired by scientific modeling domains, along with variable semantics and synthetically sampled three-dimensional data. Importantly, SURFACEBENCH incorporates geometry-aware evaluation through object-space metrics such as Chamfer and Hausdorff distances, alongside symbolic equivalence checks and regression-based errors, enabling evaluation of functional fidelity beyond algebraic syntax. The *primary contributions* of this work are as follows.

- We introduce SURFACEBENCH, a *large-scale geometry-aware benchmark for symbolic surface discovery* comprising *183 surfaces across 15 categories* and three representation paradigms. SURFACEBENCH establishes a new paradigm for equation discovery that moves beyond scalar functions to structured, multi-output, geometry-aware expressions.

- We establish a geometry-aware evaluation framework that integrates symbolic equivalence checks with object-space metrics, addressing representational non-uniqueness beyond string-based comparison. Using this protocol, we evaluated a broad spectrum of classical and LLM-guided symbolic regression methods and find that no approach consistently generalizes across representations, with exact recovery rates of only 4% for LLM-based frameworks and 6% for traditional methods. These results expose substantial headroom in structure discovery, parameter calibration, and multi-output reasoning.

- We provide an in-depth error taxonomy, decomposing *symbolic (structural)* versus *geometric (shape-level)* failure modes. Through targeted ablations, we analyze the impact of representation type, operator composition, providing concrete failure diagnostics and generalization breakdowns that offer actionable design insights for future surface reasoning methods.

## 2 Challenges and Design of SURFACEBENCH

### 2.1 Benchmark Challenges and Motivation

SURFACEBENCH is a symbolic regression benchmark designed to evaluate symbolic reasoning capabilities that are critical for scientific equation discovery. Traditional symbolic regression tasks focus on scalar mappings such as $y = f(x)$, where evaluation is limited to one-dimensional dependencies and success is measured by numerical fit. SURFACEBENCH extends this paradigm to governing equations that describe full three-dimensional surfaces, which encapsulate multi-variable dependencies, invariances, and geometric constraints. The surfaces themselves are not the end goal, but serve as a structured substrate through which the benchmark probes symbolic reasoning under coupled geometric and analytic constraints. By representing relationships as explicit, implicit, or parametric surfaces, it tests whether a model can recover equations that jointly capture variable interactions and geometric structure. This formulation moves symbolic regression beyond scalar curve fitting toward the more rigorous demands of scientific modeling, where systems are inherently multi-output, coordinate-dependent, and frequently topologically nontrivial. To systematically evaluate these capabilities, SURFACEBENCH is organized around five defining features that collectively characterize the key challenges of symbolic regression:

**(1) Multi-output coupling:** In conventional benchmarks, outputs are independent, allowing models to fit each dimension separately. SURFACEBENCH introduces multivariate targets that depend on several interacting variables—for example, $z = \sin(x^2 + y^2)$, where changes in one variable alter the curvature throughout the surface. This setup tests whether models can reason jointly over variables that together define a governing law, rather than treating them as separable regressions. This coupling expands the symbolic search space and reflects the interdependencies typical of real-world physical systems (Moyano et al., 2021).

**(2) Latent coordinate systems and topology:** Many scientific laws admit concise symbolic forms only in transformed coordinate systems such as spherical, cylindrical, or other field-aligned representations, while observed data are typically provided in Cartesian space (Champion et al., 2019). SURFACEBENCH examines whether models can infer such latent transformations that reveal underlying invariances. The benchmark also includes surfaces with nontrivial topology, including holes, folds, or disconnected components, that are best represented as implicit level sets $f(x, y, z) = 0$ or as parametric mappings $(x(u, v), y(u, v), z(u, v))$. These formulations challenge symbolic regression pipelines that assume scalar outputs and fixed templates, requiring models to infer both the algebraic law and its structural form.

**(3) Symbolic non-uniqueness:** Multiple algebraically distinct expressions can describe identical behaviors. A sphere, for example, may appear implicitly as $x^2 + y^2 + z^2 = R^2$ or parametrically as $(R \sin \phi \cos \theta, R \sin \phi \sin \theta, R \cos \phi)$. Trigonometric identities, affine transformations, and reparameterizations multiply the number of equivalent formulations, making string-level comparison unreliable. This representational non-uniqueness motivates the need for evaluation criteria that operate in geometric rather than symbolic space (Jiang et al., 2025).

**(4) Geometry-aware evaluation:** To address symbolic non-uniqueness, SURFACEBENCH evaluates predicted and reference equations through their induced geometry. Both are sampled as dense point clouds, aligned under a similarity transform, and scored using the **Chamfer Distance** (capturing mean geometric fidelity) and **Hausdorff Distance** (capturing worst-case deviation). This procedure quantifies functional equivalence directly in object space, rewarding models that recover the correct governing law regardless of symbolic form, and explicitly decoupling geometric fidelity from algebraic syntax (Fan et al., 2016).

**(5) Diversity and coverage:** SURFACEBENCH spans 15 scientifically grounded categories and 183 validated equations across explicit, implicit, and parametric formulations. This diversity ensures broad coverage of symbolic operators, compositional patterns, and topological structures, providing a rigorous and representative testbed for assessing symbolic reasoning across scientific domains.

## 2.2 Benchmark Construction Pipeline

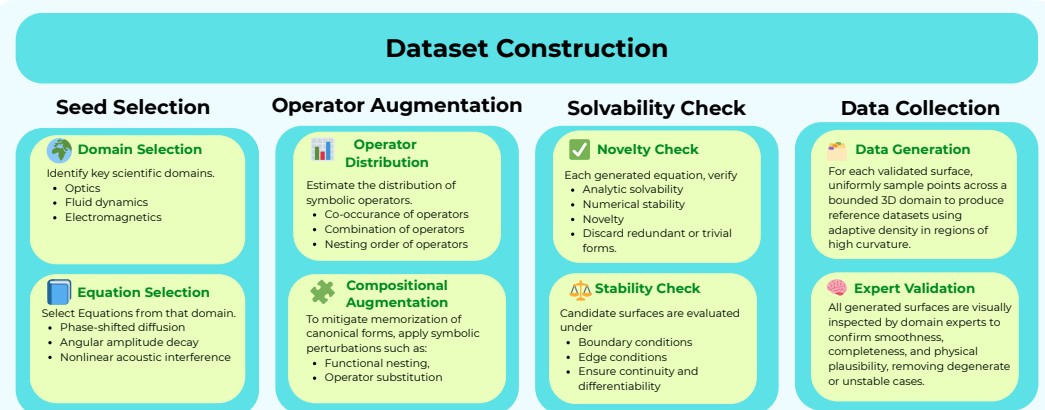

Figure 2: Dataset curation pipeline for SURFACEBENCH, ensuring a diverse set of seed equations, their transformation to discourage memorization, and rigorous validation through novelty and solvability checks.

The construction of SURFACEBENCH, as visualized in Figure 2, follows a structured multi-phase pipeline designed to ensure analytic validity, structural diversity, and robustness against memorization. **(1) Domain Selection:** We first identify scientifically motivated problem settings (such as optics, fluid dynamics, electromagnetics, materials science, and robotics) that naturally give rise to continuous 3D surfaces expressible through analytic equations, which serve as principled templates for surface construction. **(2) Equation Selection:** Within each domain, representative problems are selected and their analytic surface equations are obtained in explicit, implicit, or parametric form, providing interpretable analytic seeds for transformation. **(3) Operator Distribution:** The collected corpus is analyzed to estimate the empirical distribution of symbolic operators (e.g., trigonometric, exponential, rational, polynomial), ensuring that later augmenta-

tions reflect realistic symbolic usage patterns commonly observed in scientific modeling, rather than relying on arbitrary or statistically ungrounded operator compositions. **(4) Compositional Augmentation:** To mitigate memorization of canonical forms, we apply controlled symbolic perturbations inspired by LLM-SRBench (Shojaee et al., 2025b), including functional nesting (e.g., $\sin(x) \rightarrow \sin(x^2 + y^2)$), additive and multiplicative term blending, coordinate reparameterization (affine, polar, or spherical substitutions), and operator substitution (e.g., replacing $\sin(x)$ with $\tanh(x)$ or $(1 - e^{-x})$). These augmentations produce non-canonical yet analytically solvable variants that maintain interpretability while forcing models to reason compositionally rather than retrieve memorized templates. **(5) Novelty Check:** Each generated equation is symbolically simplified and verified for analytic solvability, numerical stability, and novelty relative to prior benchmarks (Feynman (Udrescu & Tegmark, 2020), SRBench (La Cava et al., 2021)), discarding redundant or trivial forms. **(6) Stability Check:** Candidate surfaces are evaluated under boundary and edge conditions (e.g., $x, y = 0$ or asymptotic limits) to ensure continuity, differentiability, and bounded evaluation domains. **(7) Data Generation:** For each validated surface, we uniformly sample points across a bounded 3D domain to produce reference datasets for explicit, implicit, and parametric formulations, using adaptive sampling density in regions of high curvature. **(8) Expert Validation:** Finally, all surfaces are visually inspected by domain experts to confirm smoothness, completeness, and analytic consistency, removing degenerate or unstable cases. This pipeline yields a benchmark of 183 rigorously validated surfaces with three canonical representations (explicit, implicit, and parametric), spanning 15 structural surface categories.

## 3 Experimental Setup

### 3.1 Benchmark Methods

We evaluate SURFACEBENCH (see Figure 3) using a representative suite of symbolic regression frameworks that capture both classical and LLM-driven approaches. Together, these baselines span evolutionary search, neural-guided search, reinforcement learning, and LLM-based symbolic hypothesis generation.

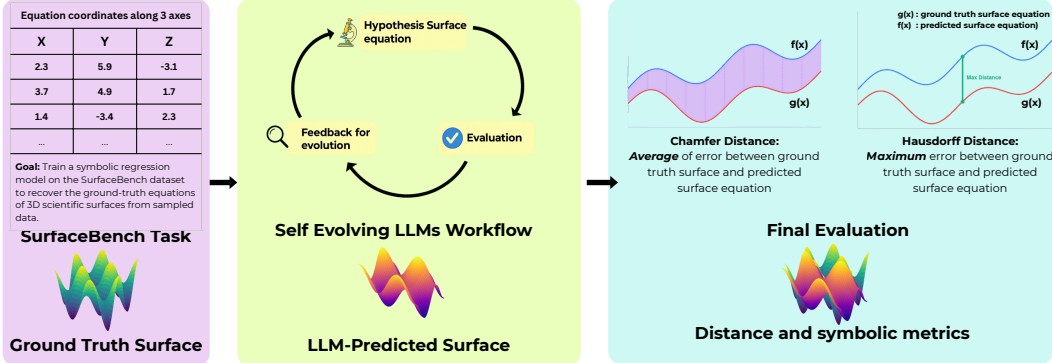

Figure 3: The SURFACEBENCH evaluation pipeline integrates symbolic and geometric metrics to assess equation recovery quality. Given sampled 3D surface data, symbolic regression frameworks, including self-evolving LLM-based methods, generate candidate symbolic expressions. These predictions are compared against the ground truth using three complementary evaluation modes: regression-style errors (NMSE), symbolic accuracy (via equivalence checks), and geometry-aware distance metrics, namely Chamfer and Hausdorff distances.

#### 3.1.1 LLM-Based Equation Discovery Methods

**LLM-SR (Shojaee et al., 2025a)** — A program-search framework that expresses candidate equations as Python function templates. It fuses the scientific prior knowledge of large language models with a multi-island evolutionary search, using data-driven feedback to refine hypotheses.

**LaSR (Grayeli et al., 2024)** — A concept-learning approach that abstracts high-level semantic descriptions of mathematical relations from previously successful equations. These concepts guide a hybrid discovery process that blends LLM-assisted search with evolutionary optimization implemented through PySR.

**SGA (Ma et al., 2024)** — A bilevel optimization method that alternates between symbolic structures generated by an LLM through discrete search and parameter fitting via PyTorch-based numerical optimization, thus coupling symbolic reasoning with continuous parameter refinement.

**OpenEvolve (Sharma, 2025)** — An open-source package implementation for the alphaevolve framework (Novikov et al., 2025). It is a general-purpose LLM–guided evolutionary framework that uses LLMs to propose transformation rules, mutation operators, or symbolic search heuristics, while fitness evaluation and selection are performed externally for the given tasks. This design decouples reasoning from optimization, allowing the LLM to serve as a flexible rule generator that evolves hypotheses efficiently.

### 3.1.2 Non-LLM-Based Equation Discovery Methods

**TPSR (Shojaee et al., 2023)** — A Transformer-driven symbolic regression model that integrates Monte Carlo Tree Search (MCTS) as a decoding strategy. By incorporating feedback and caching, it accelerates equation generation during inference.

**NeSymReS (Biggio et al., 2021)** — A neural symbolic regression model pre-trained on large synthetic corpora of equations. It maps expressions to latent embeddings and reconstructs symbolic skeletons, followed by gradient-based fitting of numerical constants.

**E2E (Kamienny et al., 2022)** — An end-to-end Transformer trained to output entire equations directly, without intermediate skeletons. Constant values are refined post-prediction using the BFGS optimization algorithm, and scalable inference is achieved through generative sampling.

**DSR (Petersen et al., 2019)** — A reinforcement-learning–based symbolic regression framework that directly optimizes expression trees, balancing exploration and parsimony via accuracy-driven reward functions.

**uDSR (Landajuela et al., 2022)** — An extension of DSR that performs variable reduction and adds linear tokens for polynomial construction. It employs large-scale pretraining while remaining rooted in genetic programming principles.

**PySR (Cranmer, 2023)** — A genetic programming engine for symbolic regression that automatically tunes hyperparameters and enforces dimensional consistency by penalizing expressions violating unit constraints.

**gplearn (Stephens, 2016)** — A scikit-learn–compatible genetic programming library using a fit/predict interface, enabling integration with existing ML workflows and hyperparameter tuning pipelines.

## 3.2 Evaluation Metrics

Evaluating symbolic regression for surface recovery poses unique challenges due to the vast hypothesis space and the existence of multiple algebraically distinct expressions that describe identical geometric manifolds. Prior equation discovery benchmarks typically rely on scalar regression metrics (e.g., NMSE) or symbolic string comparisons between candidate and ground-truth formulas. Such approaches are insufficient for surfaces, where functional equivalence is determined by geometric correspondence in object space rather than textual similarity. To address this, SURFACEBENCH introduces a domain-specific evaluation suite that integrates *geometry-aware distances*, *symbolic equivalence checks*, and *scale-invariant regression error*.

Owing to the nature of surfaces, multiple algebraically distinct expressions can produce the same three-dimensional manifold. These equivalences may arise from representational transformations, reparameterizations, or algebraic rearrangements. Conventional metrics such as normalized MSE or string based similarity would incorrectly penalize such cases. To capture equivalence in *functional* rather than *symbolic* space, SURFACEBENCH evaluates candidate and ground-truth surfaces directly in object space. Candidate and reference surfaces are uniformly sampled into dense point clouds and aligned under a similarity transform to remove translation, rotation, and scale differences. We adopt two standard object-space distances that together capture global and local geometric fidelity:

$$\text{Chamfer}(P, Q) = \frac{1}{|P|} \sum_{p \in P} \min_{q \in Q} \|p - q\|_2^2 \; + \; \frac{1}{|Q|} \sum_{q \in Q} \min_{p \in P} \|q - p\|_2^2, \tag{1}$$

$$\text{Hausdorff}(P, Q) = \max \left\{ \sup_{p \in P} \min_{q \in Q} \|p - q\|_2, \; \sup_{q \in Q} \min_{p \in P} \|q - p\|_2 \right\}. \tag{2}$$

Chamfer Distance measures the *mean geometric fidelity* between two surfaces, emphasizing smooth, global deviations and gradual warping across the shape. It provides a stable, differentiable indicator of overall shape fidelity and is robust to minor sampling noise or misalignment. Hausdorff Distance captures the *worst-case deviation*, making it sensitive to sharp discontinuities, holes, or missing components. It is characteristically less smooth and highlights rare but critical geometric failures that the Chamfer distance may overlook. Together, they characterize both distributed and localized geometric errors, thus revealing whether discrepancies stem from global distortion or isolated structural mismatches. Their joint use ensures both global accuracy and local structural integrity in evaluating equation discovery models for surfaces.

**Symbolic Accuracy.** Following LLM-SRBench (Shojaee et al., 2025b), we measure *Symbolic Accuracy* using an LLM-based equivalence check that incorporates algebraic simplifications and parameter rescalings. This provides a principled yet flexible mechanism for assessing symbolic equivalence beyond exact string matching. Additional details and the exact prompt are provided in Appendix A.1.

**Normalized Mean Squared Error (NMSE).** To maintain comparability with prior scalar-function benchmarks, we include NMSE as a regression-style measure of pointwise fit:

$$\text{NMSE} = \frac{\sum_{i=1}^{N_{\text{test}}} (\hat{y}_i - y_i)^2}{\sum_{i=1}^{N_{\text{test}}} (y_i - \bar{y})^2}.$$

## 4 Experimental Results

Table 1: Comparison of various symbolic regression methods on SurfaceBench. Performance is reported across explicit and implicit forms using Symbolic Accuracy (SA), Normalized Mean Squared Error (NMSE), Chamfer distance, and Hausdorff distance. Higher is better for SA and lower is better for the remaining metrics.

| Base LLM | Explicit | | | | Implicit | | | |
|---|---|---|---|---|---|---|---|---|
| | SA ↑ | NMSE ↓ | Chamfer ↓ | Hausdorff ↓ | SA ↑ | NMSE ↓ | Chamfer ↓ | Hausdorff ↓ |
| **SGA** | | | | | | | | |
| GPT4o-mini | **0.20** | **2.86** | 8.26 | 16.53 | **0.06** | **1.38** | **2.96** | **6.72** |
| Llama-3.1-8B | 0.10 | 3.73 | 9.82 | 18.48 | 0.05 | 1.43 | 3.01 | 7.81 |
| Qwen3-8B | 0.10 | 4.29 | **5.19** | **13.25** | 0.05 | 1.57 | 3.05 | 8.26 |
| **LaSR** | | | | | | | | |
| GPT4o-mini | **0.35** | **2.87** | 4.30 | **11.00** | 0.06 | 3.48 | 5.04 | 10.07 |
| Llama-3.1-8B | 0.30 | 3.21 | 3.68 | 14.21 | **0.10** | **2.81** | **4.67** | **9.78** |
| Qwen3-8B | 0.30 | 2.96 | **4.18** | 12.84 | 0.06 | 3.08 | 4.92 | 10.06 |
| **LLM-SR** | | | | | | | | |
| GPT4o-mini | **0.30** | 2.57 | 7.08 | **24.17** | 0.10 | **2.54** | 2.20 | **5.25** |
| Llama-3.1-8B | 0.20 | 2.62 | 7.44 | 29.29 | **0.13** | 2.74 | 3.01 | 9.05 |
| Qwen3-8B | 0.25 | **2.38** | **6.99** | 28.83 | 0.02 | 2.61 | **1.51** | 10.6 |
| **OpenEvolve** | | | | | | | | |
| GPT4o-mini | **0.50** | **0.98** | **2.69** | **4.88** | **0.12** | **0.71** | **1.85** | **4.96** |
| Llama-3.1-8B | 0.40 | 0.99 | 3.17 | 5.08 | 0.02 | 0.99 | 2.96 | 5.02 |
| Qwen3-8B | 0.40 | 1.25 | 3.23 | 5.82 | 0.04 | 0.92 | 2.35 | 5.92 |
| **Non-LLM Baselines** | | | | | | | | |
| NeSymReS | 0.20 | 0.59 | 0.84 | 1.23 | 0.05 | 0.69 | 2.45 | 5.95 |
| gplearn | 0.15 | 0.38 | 0.61 | 1.09 | 0.05 | 0.82 | 1.83 | **5.13** |
| E2E | 0.30 | 0.28 | 0.34 | 0.88 | 0.05 | 0.69 | 1.83 | 5.70 |
| DSR | 0.15 | 0.32 | 0.24 | 0.87 | 0.05 | 0.64 | 1.74 | 6.06 |
| uDSR | **0.25** | 0.25 | 0.30 | 0.88 | **0.10** | **0.61** | 1.79 | 5.72 |
| TPSR | **0.25** | 0.21 | 0.34 | 0.85 | 0.05 | 0.66 | 2.52 | 6.88 |
| PySR | **0.25** | **0.18** | **0.13** | **0.41** | 0.05 | 0.60 | **1.64** | 5.53 |

Table 1 reports results aggregated by representation (explicit and implicit) across four metrics: Symbolic Accuracy, NMSE, Chamfer, and Hausdorff. For completeness, we report that exact equation recovery (string-level match up to trivial simplifications) is rare: 4% for LLM-based frameworks and 6% for traditional SR baselines when aggregated across all SurfaceBench tasks. We do not tabulate this in Tables 1–2 because the rates are uniformly low across methods and therefore not discriminative. We observe that explicit surfaces yield the highest Symbolic Accuracy, whereas implicit surfaces achieve the lowest geometric distances. In

particular, results on explicit surfaces indicate that models often recover the correct structural family, but fail to produce geometrically tight parameterizations. We attribute this to symbolic structures that capture the correct functional family, while the subsequent parameter optimization stage remains under-refined. As a result, Chamfer and Hausdorff distances remain high despite strong Symbolic Accuracy. This exposes a pipeline gap: after structure discovery, a targeted geometric calibration step is needed to refine parameters such as scale and shift, ensuring that structural recovery translates into measurable gains in geometry-based metrics.

In contrast, results for the implicit category show the opposite pattern. Distance-driven search procedures bring the discovered surface equations closer to the ground-truth geometry even when the recovered algebraic form is not exact. This yields strong Chamfer and Hausdorff performance despite lower Symbolic Accuracy. Together, these findings highlight the structural tension (and complementarity) between geometric proximity and algebraic fidelity.

Table 2 reports the results for parametric surfaces. Parametric equations remain

Table 2: Evaluation of symbolic regression methods on parametric surface equations.

| Model | SA ↑ | NMSE ↓ | Chamfer ↓ | Hausdorff ↓ |
|---|---|---|---|---|
| **OpenEvolve** | | | | |
| GPT4o-mini | 0.07 | 0.84 | **1.22** | **3.98** |
| Llama-3.1-8B | 0.02 | 1.38 | 2.96 | 4.87 |
| Qwen3-8B | 0.04 | 1.25 | 2.35 | 4.59 |
| **Non-LLM Baseline** | | | | |
| PySR | **0.10** | **0.61** | 2.52 | 5.53 |

the most underexplored representation paradigm in symbolic regression. Among the methods we benchmark, only OpenEvolve and PySR reliably handle multiple coupled equations, i.e., multi-output regression within a single optimization pipeline. This exposes a major gap in current symbolic discovery methods: few frameworks are explicitly designed to jointly learn coupled equation systems, which is essential for modeling parametric surfaces. Finally, results on parametric surfaces show consistently strong performance across all metrics for both non-LLM baselines and the LLM-based OpenEvolve framework.

## 5 Robustness and Diagnostic Analysis

We conduct a comprehensive set of ablations to characterize the robustness and generalization behavior of these methods. As noted previously, the LaSR, SGA, and LLM-SR methods lack algorithmic support for parametric equation discovery; accordingly, we exclude the parametric category from these ablation studies.

### 5.1 Noise sensitivity

To evaluate robustness under realistic data perturbations, we conduct a noise sensitivity analysis across two representative LLM backbones: GPT-4o-mini and LLaMA-3.1-8B. We designed an experiment to systematically investigate how model performance degrades under varying levels of data corruption, simulating real-world scenarios where training data may contain measurement errors, sensor noise, or other sources of uncertainty. We select 13 representative equations spanning diverse structural families to ensure broad coverage of the symbolic hypothesis space. The experimental design employed a factorial approach with three noise levels (1%, 5%, and 10% Gaussian noise). Performance is evaluated using the Chamfer and Hausdorff distances on in-domain test data, providing complementary measures of geometric fidelity between predicted and ground truth. Robustness is reflected in the degradation slope as Gaussian noise increases from 1% to 10%. Traditional non-LLM SR methods exhibit comparatively modest changes in geometric error, suggesting that their performance is primarily constrained by systematic search and fitting limitations rather than input corruption. In contrast, LLM-based methods degrade substantially under increasing noise, indicating higher variance in symbolic hypothesis generation and greater sensitivity to perturbations in the training data. We additionally observe that Hausdorff often increases more sharply than Chamfer, implying that noise primarily manifests itself as localized worst-case defects rather than uniform surface drift. This distinction is critical: two methods may exhibit comparable average alignment (Chamfer) while differing markedly in structural brittleness (Hausdorff). Overall, the noise ablation highlights which pipelines maintain geometric fidelity under realistic uncertainty and which fail via localized breakdowns as corruption increases.

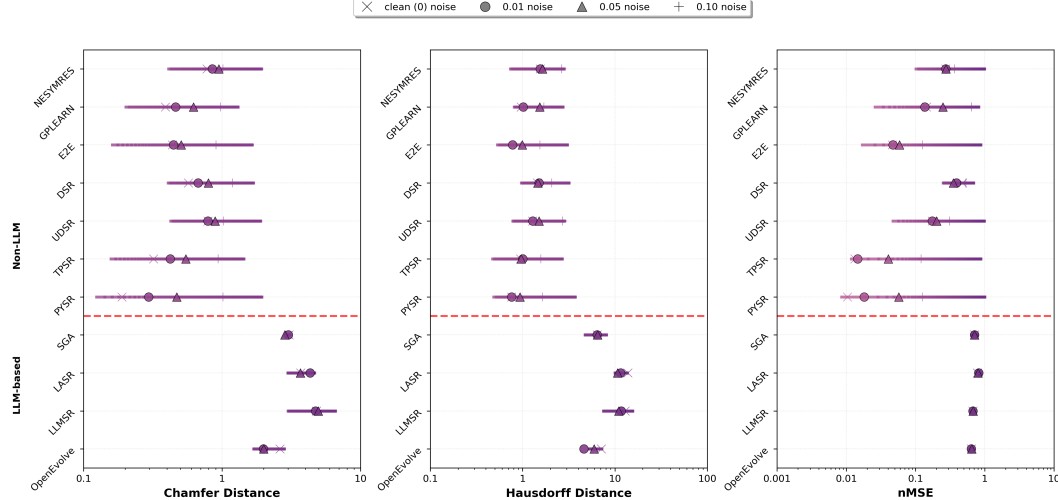

Figure 4: Noise sensitivity analysis across Chamfer Distance, Hausdorff Distance, and NMSE. Lower values indicate better performance.

## 5.2 Out-of-Domain Generalization

We define out-of-domain samples strictly as a controlled range shift in the input domain. If a model is fitted on inputs sampled from [-5,5] along each axis, OOD tests use the non-overlapping exterior bands $[-10, -5] \cup [5, 10]$. This isolates extrapolation from interpolation: models must extend learned structure beyond the training support rather than reproduce local trends. Evaluation is conducted directly in object space, using both symbolic and geometric metrics, ensuring that performance differences reflect genuine extrapolative structure preservation rather than memorized symbolic fragments.

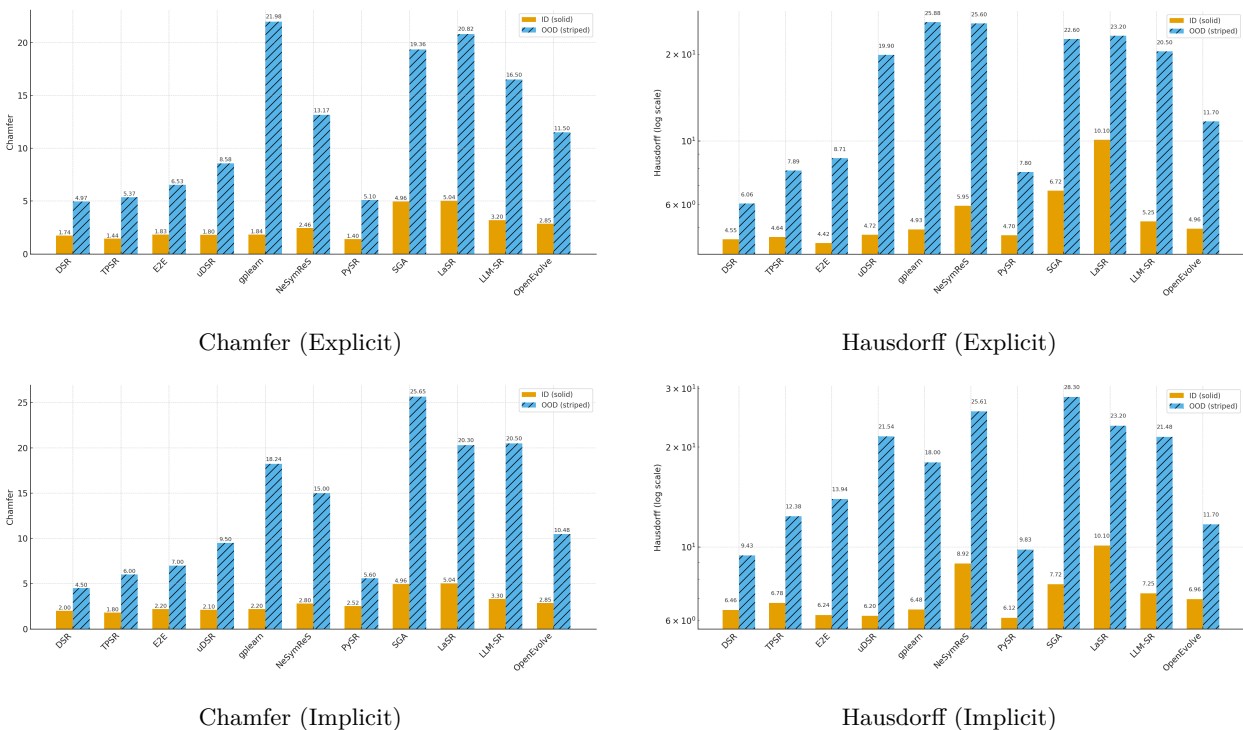

Figure 5: Chamfer and Hausdorff distance metrics for both in-domain (ID) and out-of-domain (OOD) generalization across explicit and implicit forms.

As shown in Figure 5, this range-shift protocol induces a consistent generalization gap: methods that perform strongly in-domain frequently degrade substantially under range extrapolation, indicating that many discovered expressions behave as accurate local interpolants in $[-5, 5]$ but fail to preserve correct functional behavior outside the training support. This degradation is particularly pronounced when extrapolation alters the dominant growth behavior of the expression (e.g., growth rates, asymptotes, or oscillatory behavior), where small structural or coefficient errors can rapidly amplify and translate into large geometric deviations.

We also observe that the gap is not uniform between metrics. Although Chamfer distances reflect average surface alignment, Hausdorff distances are driven by worst-case discrepancies. Thus, disproportionately larger OOD increases in Hausdorff relative to Chamfer indicate that extrapolation failures often arise from localized structural breakdowns rather than uniform global drift. Finally, differences between explicit and implicit equations provide an additional diagnostic signal: extrapolation in the explicit forms can be brittle when the recovered functional mapping has unstable out-of-range behavior, whereas implicit forms can sometimes preserve portions of the geometry even when the recovered algebraic form is imperfect.

## 5.3 Impact of Domain-Prior Prompts

To assess the utility of domain knowledge, we conduct an ablation in which domain priors are injected into prompts as lightweight structural cues describing coordinate systems such as spherical or cylindrical charts, conservation or symmetry properties, knowledge about the scientific field of the equation, such as optics, electromagnetism, etc. These prompts differ from generic discovery prompts by encoding partial structural knowledge rather than purely data-driven guidance. Although such priors can, in principle, reduce search-space ambiguity, they are not typically available in real-world experimental settings. Moreover, providing incorrect or mismatched priors often leads to severe degradation in reconstruction quality, demonstrating that overly constraining prompts can hinder rather than improve symbolic discovery.

Table 3: Comparison of performance with and without domain priors across explicit and implicit surfaces. Positive $\Delta$ indicates improvement when priors are used.

| Method | Explicit | | | | | | Implicit | | | | | |
| | Chamfer | | | Hausdorff | | | Chamfer | | | Hausdorff | | |
| | w/o priors | w/ priors | $\Delta$ | w/o priors | w/ priors | $\Delta$ | w/o priors | w/ priors | $\Delta$ | w/o priors | w/ priors | $\Delta$ |
|---|---|---|---|---|---|---|---|---|---|---|---|---|
| SGA | 7.76 | 6.22 | +1.54 | 16.09 | 12.22 | +3.87 | 3.01 | 2.76 | +0.25 | 7.60 | 6.26 | +1.34 |
| LaSR | 4.05 | 4.04 | +0.01 | 12.68 | 10.04 | +2.64 | 4.88 | 4.24 | +0.64 | 9.97 | 9.24 | +0.73 |
| LLM-SR | 7.17 | 5.77 | +1.40 | 27.43 | 20.77 | +6.66 | 2.24 | 2.07 | +0.17 | 8.30 | 5.07 | +3.23 |
| OpenEvolve | 2.93 | 1.16 | +1.77 | 4.98 | 4.16 | +0.82 | 2.41 | 1.82 | +0.59 | 4.99 | 4.82 | +0.17 |

As observed in Table 3, despite being provided with correct priors, LLM-based symbolic regression methods show only marginal improvement over their baseline performance and remain inferior to non-LLM methods. This suggests that, despite narrowing the functional search space, current LLM-based approaches do not reliably translate structural cues into improved optimization outcomes. These findings motivate a deeper failure analysis, presented in the subsequent section, to identify where such methods falter in integrating structural and geometric cues. The example prompts used for this ablation can be found in Appendix A.5.

## 5.4 Failure Analysis

Figure 6 summarizes our failure analysis, which investigates why LLM-based symbolic discovery methods underperform relative to traditional approaches. Symbolic regression requires (1) discrete structural search over symbolic expressions and (2) continuous optimization of numerical parameters. These are tightly coupled subproblems that LLM-based models attempt to solve within a single generative process, which often makes it difficult to isolate specific failure modes. To disentangle these effects, we classify errors as *search failures* and *equation fitting failures*. (i) A search failure occurs when the discovered equation includes terms from the incorrect functional family (for example, polynomials instead of trigonometric terms), indicating a breakdown in symbolic term retrieval. (ii) An equation-fitting failure, in contrast, arises when the model correctly identifies the relevant symbolic families but fails to assemble them in the correct structural order or infer accurate constants, indicating a lack of optimization quality. To ground the analysis, we evaluate LLM-SR and OpenEvolve on ten representative target equations containing trigonometric or exponential

terms; accordingly, legend entries such as Trigonometric and Exponential indicate the *target family* of each example rather than distinct error types. In Figure 6, the yellow box corresponds to search failures implying incorrect family retrieval, whereas the green box corresponds to equation-fitting failures, corresponding to correct family retrieval but poor optimization.

Non-LLM based methods perform thousands of evaluation and mutation cycles within minutes, guided by explicit loss-driven feedback. We attribute this limitation to the autoregressive generation paradigm of LLM-based approaches, which introduces computational latency and lacks explicit iterative optimization mechanisms once a candidate structure is proposed. We also observe that LLM-based methods succeed primarily when they identify the correct functional categories early in the search process. After approximately 200 iterations, it is rare for these models to meaningfully recover toward the correct structural family, suggesting limited corrective or self-refinement capability once an initial hypothesis diverges from the target function. In complete failure cases, the issue originates earlier, where the models fail to identify correct functional primitives at all, leading to collapse at the symbolic search stage rather than during optimization. While LLMs offer early-stage efficiency through linguistic priors, the absence of tightly coupled iterative optimization and feedback mechanisms limits robust convergence, especially under multi-output constraints or com-

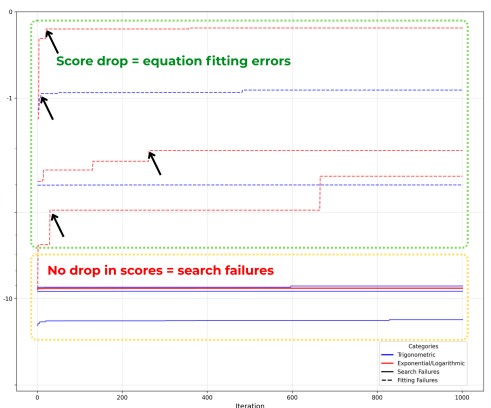

Figure 6: **Failure modes of LLM-based symbolic regression (LLM-SR, OpenEvolve).** We analyze two error modes: **(i) search space errors** (the generated expression uses primitives from an incorrect functional family) and **(ii) equation-fitting errors** (the correct families are retrieved but constants and/or composition are not optimized). Legend entries such as Trigonometric and Exponential denote the *example equation families* used in this analysis (i.e., representative targets containing trigonometric or exponential terms only). The green boxed region highlights **equation-fitting failures** , while the yellow boxed region highlights **search failures**.

plex compositional structures. These observations suggest that future LLM-based symbolic regression frameworks should incorporate stronger post-structure calibration and iterative optimization mechanisms to mitigate the dominant equation-fitting failures exposed by SurfaceBench.

# 6 Related Work

## 6.1 Symbolic Regression

Early symbolic regression (SR) research was primarily centered on genetic programming (GP), which evolves tree-structured equations using crossover and mutation, allowing discovery of interpretable analytic forms directly from data (Koza, 1992). Later variants of GP improved efficiency through regularization strategies and lexicographic tournament selection (Schmidt & Lipson, 2009; Vladislavleva et al., 2009; Agapitos et al., 2012), while others incorporated dimensional-consistency constraints (Udrescu & Tegmark, 2020). Despite success on low-dimensional problems, classical GP methods suffer from exponential search complexity and limited scalability under noisy observations or higher-order nonlinear dynamics (McConaghy, 2011; La Cava et al., 2021). In physics-guided SR, partial differential and variational approaches discover governing equations by enforcing physical constraints such as smoothness or conservation; examples include SINDy (Brunton et al., 2016) and PDE-Net (Long et al., 2018), which identify sparse terms from predefined operator libraries. Differentiable and neural SR frameworks such as DSR (Petersen et al., 2019) and NeSymReS (Biggio et al., 2021) combine neural function approximators with symbolic optimization, blending gradient-based fitting with discrete equation search to improve sample efficiency and search stability. Recently, large language models (LLMs) have been incorporated into SR pipelines, with approaches such as LLM-SR (Shojaee et al., 2025a), LaSR (Grayeli et al., 2024), and OpenEvolve (Sharma, 2025). These methods use LLMs to generate symbolic hypotheses, refine them using tool-assisted feedback, and apply self-reflective reasoning loops. While they exploit pretrained scientific priors to propose meaningful candidate expressions, these meth-

ods continue to struggle with coupled multi-equation systems, implicit and parametric representations, and topology-aware surface reconstruction.

## 6.2 Benchmarking Symbolic Regression

Early benchmarks such as AI Feynman (Udrescu & Tegmark, 2020), SRSD (Matsubara et al., 2024), and SR-Bench (La Cava et al., 2021) have been widely used to evaluate symbolic regression methods. However, these benchmarks are vulnerable to memorization, where LLMs often reproduce canonical formulas directly rather than reasoning from data (Shojaee et al., 2025b). To address this, LLM-SRBench (Shojaee et al., 2025b) introduced a large-scale set of synthetic equations specifically designed to mitigate memorization. However, LLM-SRBench remains restricted to scalar mappings and evaluates discovery quality through algebraic fit alone, without accounting for geometric equivalence or representational non-uniqueness. Additionally, its tasks are structural variations of a limited set of formula families, offering narrow coverage of the compositional and multi-output challenges encountered in scientific modeling. SURFACEBENCH extends beyond this paradigm by targeting multi-output surface equations across three representation types and introducing geometry-aware evaluation that captures functional fidelity in object space.

## 6.3 Surface Equation Discovery

The task of recovering closed-form equations that define 3D surfaces has a long history across computer graphics, geometry processing, and learning-based methods. Early work focused on fitting smooth implicit functions to point clouds using radial basis functions (Carr et al., 2001) or solving Poisson-type PDEs to generate indicator fields aligned with input normals (Kazhdan et al., 2006; Kazhdan & Hoppe, 2013). These methods yield robust reconstructions, but the resulting equations are either constrained to a predefined basis or lack closed-form interpretability. Neural implicit models such as DeepSDF (Park et al., 2019) and Occupancy Networks (Mescheder et al., 2019) generalize this idea by learning continuous fields whose level sets define geometry, achieving high fidelity but producing non-symbolic, black-box representations. Hybrid methods such as AtlasNet (Groueix et al., 2018) learn parametric surface patches, offering some structural insight but still lacking explicit symbolic representations amenable to equation-level reasoning. Collectively, these three lines of work reveal a gap at their intersection that SURFACEBENCH addresses directly: symbolic regression methods have not been evaluated on surface-level discovery tasks, existing SR benchmarks remain restricted to scalar mappings, and surface reconstruction methods have not pursued symbolic representations.

# 7 Conclusion

We introduce SURFACEBENCH, the first geometry-aware benchmark for LLM-driven *symbolic discovery of 3D surfaces*, comprising *183* tasks across *15* scientifically grounded categories and three representation paradigms: *explicit*, *implicit*, and *parametric*. Beyond scale, SURFACEBENCH redefines evaluation for equation discovery by integrating symbolic equivalence checks with object-space geometric metrics (Chamfer and Hausdorff distances) and regression-based error measures into a unified protocol that accounts for representational non-uniqueness and structural ambiguity. Unlike prior symbolic regression datasets that emphasize scalar or low-dimensional mappings, SURFACEBENCH targets surface-level discovery, where multi-variable coupling, coordinate transformations, and geometric structure must be inferred directly from data. The benchmark consists of analytically constructed, science-inspired surfaces designed to stress symbolic composition, multi-output coupling, topology, and geometry-aware reasoning under structural complexity.

Our empirical study across evolutionary, neural, and LLM-driven discovery frameworks reveals a consistent pattern: although modern methods can occasionally recover correct functional families, none demonstrate robust performance across representation types or evaluation regimes. In particular, LLM-based approaches exhibit strong early structural priors but struggle with iterative refinement and parameter calibration, highlighting a fundamental tension between autoregressive symbolic generation and continuous optimization. Performance remains far from saturation, underscoring the need for tighter integration between discrete structure search, geometric alignment, multi-equation coupling, and differentiable parameter estimation. By releasing both the curated dataset and the accompanying geometry-aware evaluation pipeline, we aim to establish SURFACEBENCH as a community benchmark for symbolic surface discovery. We anticipate that it will catalyze advances at the intersection of symbolic regression, geometric learning, and scientific induction, and serve as a standardized platform for assessing structure-aware compositional reasoning in high-dimensional equation discovery.

## 8    Acknowledgments

This research was partially supported by the U.S. National Science Foundation (NSF) under Grant No. 2416728 and Autodesk Research.

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

# A    Appendix

## A.1    Symbolic Accuracy

In addition to geometry-based metrics, we also report **Symbolic Accuracy** to provide a complementary view of equation discovery performance. We adopt the evaluation methodology introduced in (Shojaee et al., 2025b), which leverages GPT-4o as an automated evaluator to assess the mathematical equivalence between the predicted and ground-truth hypotheses. Traditional exact-match metrics (e.g., recovery rate, tree edit distance) are insufficient in our setting, as many surface equations admit multiple algebraically equivalent representations. The LLM-based evaluator provides a more flexible and semantically meaningful assessment of symbolic equivalence, operating in diverse formats (strings, trees, and executable forms).

We follow the same preprocessing pipeline as (Shojaee et al., 2025b), including normalization of constants and removal of auxiliary information, and rely on GPT-4o's judgement of equivalence. This ensures comparability with prior symbolic regression benchmarks, while complementing our primary focus on geometric fidelity. As shown in Figure 7, the symbolic evaluation provides a complementary view of the performance of the discovery of equations.

## A.2    Memorization Resistance

To quantify the novelty of SurfaceBench relative to existing symbolic regression benchmarks, we compare each SurfaceBench equation against all equations in AI-Feynman, SRBench, and LLM-SRBench using two complementary similarity measures.

**Operator-set similarity** measures overlap between the sets of symbolic operators appearing in two expressions (e.g., $\{+, \times, \sin, \exp, \text{pow}\}$), capturing commonality in operator usage while ignoring compositional structure. **AST-signature similarity** is computed by extracting a normalized abstract syntax tree (AST) signature that summarizes structural properties of an expression, including operator composition patterns, function types, tree depth, and node counts. Similarity between signatures is then computed using a normalized similarity score.

For each SurfaceBench equation, we compute both metrics against all equations in each prior benchmark and report the *maximum* similarity observed under each metric. Using this protocol, we observe consistently low overlap with existing benchmarks: operator-set similarity has a mean of 0.284 across SurfaceBench, with

---

**Prompt for Symbolic Accuracy Evaluation**

**Question:** Given the ground truth mathematical expression **A** and the hypothesis **B**, determine if there exist any constant parameter values that would make the hypothesis equivalent to the given ground truth expression.

Let's think step by step. Explain your reasoning and then provide the final answer as:

- **(A):** `sqrt(q1/(E*epsilon))/(2*sqrt(pi))`

- **(B):** Hypothesis as Program

**LLM (GPT-4o) Judgement:**
**Reasoning:** "The expressions can match if params[0] * params[1] = 1 and params[2] = 1, as this aligns both the scalar and constant factors appropriately."
**Answer:** Yes

---

Figure 7: Prompt used for symbolic accuracy evaluation.

no equations exceeding 0.7, and AST-signature similarity has a mean of 0.407, with no equations exceeding 0.65. No exact or near-duplicate equations are observed.

Table 4 summarizes the average similarities by benchmark. These results indicate that SurfaceBench equations are structurally distinct from those of prior symbolic regression benchmarks, supporting that the benchmark resists direct memorization and near-match retrieval by LLM-based methods.

Table 4: Average operator-set and AST-signature similarity between SurfaceBench equations and prior symbolic regression benchmarks.

| Benchmark | Operator Similarity | AST Similarity |
|---|---|---|
| AI-Feynman | 0.118 | 0.191 |
| SRBench | 0.138 | 0.272 |
| LLM-SRBench | 0.284 | 0.407 |

### A.3   Implementation Details

For a comprehensive evaluation, we implement three state-of-the-art LLM-guided scientific equation discovery baselines, each tested on the SURFACEBENCH datasets using three different LLM backbones: open-source models (Llama-3.1-8B-Instruct), (Qwen3-8B), and a closed-source model (GPT-4o-mini).

### A.3.1   Parameters

Table 5 presents the key implementation details for each discovery agentic method. We adopt most of the hyperparameters from the original implementation for these methods. We have only changed some hyperparameters in different baselines that affect the number of LLM calls in the search framework. This is to make sure we have a fair comparison across baseline discovery frameworks with the same access budget to LLMs. In our experiments, all baseline frameworks have 1k calls to LLMs (per problem) through the discovery process and an equivalent number of calls to the Non-LLM method.

Table 5: Implementation details of LLM-based scientific equation discovery methods.

| Method | Parameters |
|---|---|
| OpenEvolve | Temperature $\tau = 0.8$
5 equation program hypotheses sampled from LLM for initial prompt
No access to data for data-driven refinement
Time limit $T = 30$s per program hypothesis execution,
BFGS optimizer from Scipy for parameter optimization of equation skeletons |
| SGA | PyTorch-based implementation of model and `torch.nn.Module` class
Mean square error loss for data-driven feedback in agentic search
Adam optimizer in PyTorch for differential parameter optimization of equation skeletons |
| LaSR | Iterations $= 25$
Cycles per iteration $= 550$
Populations $= 10$
Population size $= 33$
Maximum size $= 30$
Operators: $+, *, -, /, \wedge$, exp, log, sqrt, sin, cos, tan, cosh
LLM weights: llm_mutate $=0.005$, llm_crossover $=0.005$, llm_gen_random $=0.005$
Top-$K = 20$ concepts from library
Default configuration of PySR for parameter optimization |
| LLM-SR | Temperature $\tau = 0.8$
Batch size $b = 4$ equation programs per prompt
$e = 4$ parallel evaluators
Time limit $T = 30$s per program hypothesis,
Memory limit M $= 2$GB
$m = 10$ islands for population diversity through search
$k = 2$ in-context examples per prompt
Maximum 10 parameters per equation skeleton
BFGS optimizer from Scipy for parameter optimization of equation skeletons |

### A.4  Task Pipelines

**Implicit surfaces ($f(x, y, z) = 0$).**  We cast implicit surface discovery as scalar regression. We sample 3D points $(x, y, z)$ from a fixed benchmark domain and use ground-truth values $f(x, y, z)$ as regression targets. Each symbolic regression baseline produces a scalar expression $\hat{f}(x, y, z)$ by minimizing MSE on these samples. We then extract the predicted surface as the zero-level set $\hat{f}(x, y, z) = 0$ via dense sampling (e.g., grid-based extraction) and evaluate geometric fidelity against the ground-truth surface using Chamfer and Hausdorff distances.

**Parametric surfaces ($(x(u, v), y(u, v), z(u, v))$).**  Parametric surfaces require discovering multiple coupled coordinate functions, which most traditional SR baselines do not reliably support. We therefore evaluate only PySR and OpenEvolve on parametric tasks. OpenEvolve natively supports multi-equation programs and jointly optimizes $(\hat{x}(u, v), \hat{y}(u, v), \hat{z}(u, v))$. PySR is applied by fitting $\hat{x}(u, v)$, $\hat{y}(u, v)$, and $\hat{z}(u, v)$ independently from identical $(u, v)$ samples and operator sets. Other SR baselines are excluded due to the lack of stable multi-output support.

### A.5  Prompt templates for domain informed ablation

Section 5.3 describes explicit and implicit domain priors qualitatively. For reproducibility, we provide (i) the **exact prompt text** used for domain priors, (ii) the **exact equation** (reported *outside* the prompt), and (iii) the **structured inputs** provided alongside the prompt: coordinate hints, domain bounds, domain label, and symmetry hints.

**Inputs provided to the LLM (in addition to the prompt text).**  For each instance, we provide: `[DOMAIN_LABEL]`, `[DOMAIN_BOUNDS]`, `[COORD_HINTS]`, `[SYMMETRY_HINTS]`, and `[SAMPLES]`. For explicit surfaces, `[SAMPLES]` is a JSON array of $(x, y, z)$ tuples. For implicit surfaces, `[SAMPLES]` is provided as `[SAMPLES_ON]` (on-surface $(x, y, z)$ points) and optionally `[SAMPLES_NEAR]` (near-surface $(x, y, z, \pm 1)$ tuples). The prompt text below is unchanged.

**Example E1: `Parabola_Sine_Piecewise`:**

$$z = \begin{cases} x^2, & x < 0, \\ \sin(y), & x \geq 0. \end{cases}$$

**Structured inputs (provided alongside the prompt text)**

```
[DOMAIN_LABEL] = Parabola_Sine_Piecewise
[DOMAIN_BOUNDS] = {
    "x":[-5,5],
    "y":[-5,5]
}
[COORD_HINTS] = [
    "cartesian",
    "piecewise split on x",
    "allow trig in y"
]
[SYMMETRY_HINTS] = [
    "y-axis symmetry"
]
[SAMPLES] = JSON array of (x,y,z) tuples
```

**Domain-prior prompt text**

The surface exhibits a distinct transition in behavior based on the x-axis, with a parabolic profile for negative x-values and a sinusoidal form for non-negative x-values. It is closed and lacks voids or tunnels, maintaining a continuous structure without self-intersections. The surface demonstrates symmetry about the y-axis, with periodic features along the y-axis in the sinusoidal region. Anisotropic characteristics are evident, with curvature concentration varying significantly between the two defined regions.

Figure 8: Prompt E1 for the hybrid piecewise surface `Parabola_Sine_Piecewise` ($z = x^2$ for $x < 0$, $z = \sin(y)$ for $x \geq 0$). The figure presents the full prompt construction for a *hybrid multi-modal symbolic surface* whose governing expression changes across a piecewise boundary at $x = 0$. As in our prompt design, the specification combines (i) a *structured input block* and (ii) a *domain-prior natural-language description*.

**Example I1 (implicit): `Quartic_Sphere`.**

$$x^4 + y^4 + z^4 = 1.$$



**Structured inputs (provided alongside the prompt text)**

```
[DOMAIN_LABEL] = Quartic_Sphere
[DOMAIN_BOUNDS] = {
    "x":[-5,5],
    "y":[-5,5],
    "z":[-5,5]
}
[COORD_HINTS] = [
    "implicit: f(x,y,z)=0",
    "polynomial basis: xⁿ, yⁿ, zⁿ"
]
[SYMMETRY_HINTS] = [
    "sign-flip invariance",
    "permutation invariance"
]
[SAMPLES] = JSON array of (x,y,z) on-surface points
```

**Domain-prior prompt text**

This surface is closed and bounded, exhibiting a smooth topology without voids or tunnels. It possesses symmetrical properties about all principal axes, maintaining invariance under sign-flip and permutation. The surface features lobes and pinches, with curvature concentration evident in certain regions. Anisotropic scaling trends are observed as one moves away from the origin, affecting the surface's profile along the axes.



Figure 9: Prompt I1 for the implicit surface `Quartic_Sphere` ($x^4 + y^4 + z^4 = 1$). The figure shows the complete prompt specification used for this example, combining (i) a *structured input block* and (ii) a *domain-prior natural-language description.*

## A.6   Dataset Statistics

The Table 6 details the split of each domain by category.

Table 6: **SurfaceBench Dataset Statistics.** The benchmark spans 183 analytically valid, science-inspired symbolic surface equations across 15 structural categories and 3 representation types (explicit, implicit, and parametric). Each surface includes 5k training, 500 test, and 500 out-of-domain (OOD) samples, enabling evaluation across symbolic, geometric, and regression modalities. While these surface equations are inspired by forms commonly encountered in scientific modeling, SurfaceBench focuses on analytically constructed surfaces designed to probe symbolic composition, multi-output coupling, and geometry-aware reasoning rather than reproducing full governing physical laws.

| Category | Count | Highlights |
|---|---|---|
| Nonlinear Analytic Composition Surfaces | 11 | Nested analytic expressions with trigonometric, hyperbolic, and exponential operators |
| Piecewise Surfaces | 10 | Conditional logic, branching behavior, and discontinuities |
| Mixed Transcendental Analytic Surfaces | 9 | Combinations of trigonometric, exponential, and logarithmic operators with symbolic constants |
| Conditional Multi-Regime Surfaces | 9 | Piecewise definitions with distinct operator families across regimes |
| Oscillatory Composite Surfaces | 11 | Nested and summed trigonometric compositions with varying frequencies |
| Trigonometric–Exponential Composition Surfaces | 10 | Smooth bounded surfaces combining trigonometric and exponential operators |
| Multi-Operator Composite Surfaces | 10 | Mixed polynomial, trigonometric, hyperbolic, and exponential compositions |
| Elementary Bivariate Surfaces | 10 | Standard two-variable analytic functions with simple operator structure |
| Discrete Integer-Grid Surfaces | 10 | Surfaces defined on integer indices using modulo, sign, and continuous operators |
| Nonlinear Coupled Surfaces | 10 | Variable coupling through products, nested functions, and nonlinear mixing |
| Exponentially-Modulated Surfaces | 10 | Trigonometric functions modulated by exponential or Gaussian envelopes |
| Radially Decaying Surfaces | 10 | Spatially localized functions with Gaussian, sigmoidal, or bounded decay profiles |
| Polynomial Transcendental Mixtures | 9 | Polynomial terms combined with transcendental functions |
| Implicit Surfaces | 24 | High-order polynomial and transcendental implicit equations $f(x,y,z) = 0$ |
| Parametric Multi-Output Surfaces | 30 | Multi-output equations in latent coordinates $(x(u,v), y(u,v), z(u,v))$ |

## A.7   SurfaceBench equations for each scientific categories

Table 7: Exact equations used in SURFACEBENCH after applying transformations. This table provides the equations by categories of scientific domains.

| Category | ID | Equations |
|---|---|---|
| Nonlinear Analytic Composition Surfaces | NACS1 | $\sin(x^2 + y^2)/1 + x^2 + y^2$ |
| | NACS2 | $x^2 - y^2/1 + x^2 + y^2$ |
| | NACS3 | $\mathrm{atan2}\,(x, y)\,\exp\big(-(x^2 + y^2)\big)$ |
| | NACS4 | $\tanh\big(\sin(xy)\big)$ |
| | NACS5 | $\log\big(1 + x^2 + y^2\big)\,\sin(x - y)$ |
| | NACS6 | $\exp\big(\sin(x^2 + y^2)\big)$ |
| | NACS7 | $\cos(x^2 + y)/1 + |xy|$ |
| | NACS8 | $\sinh(xy)\,\exp(-y^2)$ |
| | NACS9 | $\sin\big(\sqrt{x^2 + y^2}\big)/\log(1 + x^2)$ |
| | NACS10 | $x\,\exp\big(-x^2 - y^2\big)\,\cos y$ |
| | NACS11 | $\sin(xy)/1 + x^2 + y^2$ |
| Piecewise Surfaces | PDS1 | $x^2$ if $x < y$ else $y^2$ |
| | PDS2 | $\sin(x)$ if $x < 0$ else $\exp(y)$ |
| | PDS3 | $xy$ if $xy > 0$ else $-xy$ |
| | PDS4 | $x^2 + y^2$ if $x < y$ else $x^2 - y^2$ |
| | PDS5 | $\cos(x)$ if $|x| < 1$ else $\exp(-y^2)$ |
| | PDS6 | $x^3$ if $y > 0$ else $-y^3$ |
| | PDS7 | $|x - y| + \sin(x)$ |
| | PDS8 | $\sin(x + y)$ if $x^2 + y^2 < 1$ else $0$ |
| | PDS9 | $\tanh(x)$ if $x > y$ else $\cos(y)$ |
| | PDS10 | $xy$ if $|x - y| < 0.5$ else $\sin(x - y)$ |
| Mixed Transcendental Analytic Surfaces | MTAS1 | $\sin(x) + \exp(-y^2)$ |
| | MTAS2 | $\tanh(xy) + x^2$ |
| | MTAS3 | $\exp(-x^2 - y^2) + \cos(3x)$ |
| | MTAS4 | $\alpha \sin(\beta x) + \gamma \log(1 + y^2)$ |
| | MTAS5 | $\sinh(x) - \tanh(y)$ |
| | MTAS6 | $\sin(x^2 + y^2)\,\exp(-\sqrt{x^2 + y^2})$ |

*Table 7 - continued from previous page*

| Category | ID | Equations |
|---|---|---|
| | MTAS7 | $\tanh(x) \log(1 + y^2)$ |
| | MTAS8 | $\cos(xy) + \exp(-x^2 + y)$ |
| | MTAS9 | $\beta \cos(x) + \gamma \sin(y^2)$ |
| Conditional Multi-Regime Surfaces | CMRS1 | $x^2$ if $x < 0$ else $\sin(y)$ |
| | CMRS2 | $\log(1 + |x|)$ if $y < 0$ else $\exp(-y^2)$ |
| | CMRS3 | $x^2 + \sin(y)$ if $xy > 0$ else $(-x^2 - \cos y)$ |
| | CMRS4 | $\tanh(x - y)$ if $x > y$ else $0$ |
| | CMRS5 | $|xy| + \sin(x - y)$ |
| | CMRS6 | $x^2$ if $y > 0$ else $\cos(y^2)$ |
| | CMRS7 | $\sin(xy)$ if $x^2 + y^2 < 1$ else $\log(1 + x^2)$ |
| | CMRS8 | $\tanh(x + y)$ if $xy < 0$ else $\sin(x - y)$ |
| | CMRS9 | $x$ if $x > y$ else $y^2 + \sin(x)$ |
| Oscillatory Composite Surfaces | OCS1 | $\sin(5x) \cos(5y)$ |
| | OCS2 | $\cos(x^2 y^2) + 0.2 \sin(5\sqrt{|x| + |y|})$ |
| | OCS3 | $\sin(xy) + 0.5 \sin(3x + 5y)$ |
| | OCS4 | $e^{-0.1(x^2 + y^2)} \sin(xy)$ |
| | OCS5 | $\sin(x^3 + y^3)$ |
| | OCS6 | $xy \cos(\sqrt{x^2 + y^2})$ |
| | OCS7 | $\sin(2^x x) \cos(2^y y)$ |
| | OCS8 | $e^{-|x-y|} \sin(3(x + y))$ |
| | OCS9 | $\sum_{i=1}^{4} \dfrac{\sin(2^i x)}{i}$ |
| | OCS10 | $\tanh(xy) \cos(\sqrt{x^2 + y^2})$ |
| | OCS11 | $\sin(2x) + \alpha \exp(-y^2)$ |
| Trigonometric- Exponential Composition Surfaces | TECS1 | $\sin(\sqrt{x^2 + y^2})$ |
| | TECS2 | $\exp(-x^2 - y^2) \cos(3x)$ |
| | TECS3 | $\tanh(x + y) \sin(xy)$ |

*Table 7 - continued from previous page*

| Category | ID | Equations |
|---|---|---|
| | TECS4 | $\log(1 + x^2 + y^2)\,\cos(xy)$ |
| | TECS5 | $x^2 + y^2 - \sin(2x + 2y)$ |
| | TECS6 | $\cos(2x)\,\cos(2y)$ |
| | TECS7 | $\sin(x^2 + y^2)/1 + x^2 + y^2$ |
| | TECS8 | $\tanh(x^2 - y^2)$ |
| | TECS9 | $\exp\big(-|xy|\big)\,\sin(x + y)$ |
| | TECS10 | $\cos(xy) + 0.1\,(x^2 + y^2)$ |
| Multi-Operator Composite Surfaces | MOCS1 | $\log\big(1 + x^2 + y^2\big)\,\cos(x - y)$ |
| | MOCS2 | $\sin(x) + \cos(y)/1 + x^2 + y^2$ |
| | MOCS3 | $e^{-0.1|xy|}\,\tanh(x + y)$ |
| | MOCS4 | $x^2 y - y^2/1 + x^2$ |
| | MOCS5 | $\sqrt{1 + x^2 + y^2}\,\sin(xy)$ |
| | MOCS6 | $e^x + e^{-y}/1 + |x - y|$ |
| | MOCS7 | $\begin{cases} x^2 + y^2, & \text{if } x + y < 0, \\ \sin(x + y), & \text{if } x + y \geq 0 \end{cases}$ |
| | MOCS8 | $\cos\big(\sqrt{x^2 + y^2}\big)/1 + e^{-xy}$ |
| | MOCS9 | $\sinh(x^2 - y^2)\,e^{-0.1(x+y)^2}$ |
| | MOCS10 | $\arctan(xy) + 0.2\,e^{-x^2 - y^2}$ |
| Elementary Bivariate Surfaces | EBS1 | $x^2 + y^2$ |
| | EBS2 | $\sin(x)\,\cos(y)$ |
| | EBS3 | $\exp(-x^2 - y^2)$ |
| | EBS4 | $x\,y$ |
| | EBS5 | $\tanh(x + y)$ |
| | EBS6 | $\cos(x^2 + y^2)$ |
| | EBS7 | $\log(1 + x^2 + y^2)$ |
| | EBS8 | $x^2 - y^2$ |
| | EBS9 | $\sin(x\,y)$ |
| | EBS10 | $\exp\big(-|x - y|\big)$ |

*Table 7 - continued from previous page*

| Category | ID | Equations |
|---|---|---|
| Discrete Integer-Grid Surfaces | DIGS1 | $\sin(i) + \cos(j)$ |
| | DIGS2 | $(-1)^i (-1)^j$ |
| | DIGS3 | $\mod(i, 3) + \mod(j, 2)$ |
| | DIGS4 | $\left\lfloor \sqrt{i^2 + j^2} \right\rfloor$ |
| | DIGS5 | $\sin(ij) + i - j$ |
| | DIGS6 | $\cos(i + j)$ |
| | DIGS7 | $\mod(i^2 + j^2, 5)$ |
| | DIGS8 | $\tanh(i - j)$ |
| | DIGS9 | $\left\lfloor \sin(i^2 + j^2) \right\rfloor$ |
| | DIGS10 | $\mod(ij, 4)$ |
| Nonlinear Coupled Surfaces | NCS1 | $\cosh\big(0.1(x - y)\big) - \cos\big(0.5(x + y)\big)$ |
| | NCS2 | $e^{-0.05(x^2 + y^2)} (x^2 - y) \cos(y)$ |
| | NCS3 | $\log(1 + x^2) \sin(y) - \log(1 + y^2) \cos(x)$ |
| | NCS4 | $\sqrt{1 + 0.1(x^2 + y^2)} \, \sin\big(0.5(x - y)\big)$ |
| | NCS5 | $\tanh\big(0.2(x^2 - y^2)\big)$ |
| | NCS6 | $0.3(xy) - 0.2 \sin(x + y) \, e^{-0.05(x^2 + y^2)}$ |
| | NCS7 | $x^2 \sin(y)/1 + 0.2y^2$ |
| | NCS8 | $\sinh(0.2x) \, e^{-0.1y^2}$ |
| | NCS9 | $\arctan(xy) - 0.3 \sin(x - y)$ |
| | NCS10 | $\arctan(xy) + \sin(x + y)$ |
| Exponentially Modulated Surfaces | EMTS1 | $3 \, e^{-0.05(x^2 + y^2)} \cos(0.2xy) + 0.1x$ |
| | EMTS2 | $2.2 \sin(0.3x + 0.2y) \left(1 - e^{-0.1x^2}\right)$ |
| | EMTS3 | $1.8 \cos(0.4xy) \, e^{-0.1x^2} + 0.3y^2$ |
| | EMTS4 | $2 \sin(0.7x) \, e^{-0.05y^2} + 0.5xy$ |
| | EMTS5 | $3 \left(1 - e^{-0.15x^2}\right) \cos(0.3y) + 0.2x$ |
| | EMTS6 | $2.5 \tanh(0.2xy) + 0.4 \sin(0.5x + y)$ |
| | EMTS7 | $1.5 e^{-0.1(x^2 + y^2)} \sin(0.6x) + 0.3y$ |

*Table 7 - continued from previous page*

| Category | ID | Equations |
|---|---|---|
| | EMTS8 | $4\cos(0.4x)\left(1 - e^{-0.05y^2}\right) + 0.1x^2$ |
| | EMTS9 | $2x^2 e^{-0.2|y|} + 1.5\sin(0.3xy)$ |
| | EMTS10 | $3\sin(0.5x)\,e^{-0.1y^2} + 0.2xy\cos(y)$ |
| Radially-Decaying Surfaces | LRDS1 | $e^{-0.8(x^2+y^2)}$ |
| | LRDS2 | $x^2\,e^{-(x^2+y^2)}$ |
| | LRDS3 | $(x^2 + y^2)\,e^{-0.9(x^2+y^2)}$ |
| | LRDS4 | $e^{-0.4(x^2+y^2)}\left(1.1 + \cos(5x)\right)$ |
| | LRDS5 | $\left(\cos(1.5x)\cos(1.5y)\right)^2$ |
| | LRDS6 | $\sin^2\!\left(3\arctan(\tfrac{y}{x})\right)e^{-\sqrt{x^2+y^2}}$ |
| | LRDS7 | $1 - \tanh(x^2 + y^2 - 4)$ |
| | LRDS8 | $(x^2 - y^2)^2\,e^{-0.7(x^2+y^2)}$ |
| | LRDS9 | $\sin^2(x + y)\,\cos^2(x - y)$ |
| | LRDS10 | $1 + x^2/1 + (x^2 + y^2)^2$ |
| Polynomial Transcendental Mixtures | PTM1 | $x^3 + y^3 - 3xy + \sin(x)$ |
| | PTM2 | $\log\!\left(1 + x^2 + y^2\right) - \tanh(x - y)$ |
| | PTM3 | $e^{-x^2-y^2}\,\sin(2x + y)$ |
| | PTM4 | $\arctan(x) + \arctan(y)$ |
| | PTM5 | $\sin(x)\cos(y) + 0.1\,xy$ |
| | PTM6 | $3\sin(0.4x)\,e^{-0.05y^2} + 0.2\,xy$ |
| | PTM7 | $2\sin(x + y)\,e^{-0.5x^2} + y^2$ |
| | PTM8 | $\tanh(xy) + 0.5\,\sin(0.5x)\,y$ |
| | PTM9 | $1.5\,x^2\cos(0.2y) + 0.3\,e^{-0.1x^2}$ |
| High-Degree Implicit Surfaces | HDIS1 | $x^3 + y^3 + z^3 - 3xyz = 0$ |
| | HDIS2 | $x^3y + y^3z + z^3x = 0$ |
| | HDIS3 | $x^5 + y^5 + z^5 - xyz = 0$ |
| | HDIS4 | $x^4y - z^6 + \sin(xz) = 1$ |
| | HDIS5 | $z^5 + x^3y^4 - e^y = 0$ |

*Table 7 - continued from previous page*

| Category | ID | Equations |
|---|---|---|
| | HDIS6 | $x^6 - y^4 z^2 + \tan(z) = 2$ |
| | HDIS7 | $z^3 + x^4 y^3 - \cos(x) = -1$ |
| | HDIS8 | $x^3 y^2 - z^5 + \sin(yz) = 0$ |
| | HDIS9 | $x^2 y^3 z - z^4 + \sin(x) = -1$ |
| | HDIS10 | $z^3 + x^5 y - e^z + xy^2 = 0$ |
| | HDIS11 | $x^4 - y^2 z^5 + \tan(z) = 2$ |
| | HDIS12 | $z^5 + x^3 y^4 - \cos(y) = 1$ |
| | HDIS13 | $x^6 y^2 - z^3 + e^x = 0$ |
| | HDIS14 | $z^4 - x^4 y + \sin(xz) = -2$ |
| | HDIS15 | $x^3 + y^4 z^2 - e^y + \cos(x) = 1$ |
| | HDIS16 | $x^5 y - z^4 + \sin(yz) = 2$ |
| | HDIS17 | $z^3 - x^3 y + e^z + 2xy = -1$ |
| | HDIS18 | $x^4 + y^5 z - \cos(xz) = 0$ |
| | HDIS19 | $x^6 - y^3 z^2 + \tan(z) = 2$ |
| | HDIS20 | $x^2 y^2 z - z^5 + \sin(xz) = 1$ |
| | HDIS21 | $z^3 + x^4 y - 2e^z + xy^2 = 0$ |
| | HDIS22 | $x^5 - y^2 z^3 + \cos(xy) = -1$ |
| | HDIS23 | $x^3 + y^4 z - \tan(x) + 1 = 0$ |
| | HDIS24 | $z^5 + x^3 y^2 - 2z^2 x + \sin(y) = 1$ |
| Parametric Multi-Output Surfaces | PMOS1 | $(\sinh(u/5),\ \cosh(uv/10),\ \sin(u+v)\log(1+v^2))$ |
| | PMOS2 | $(u^2 \cos v,\ v^2 \sin u,\ \tanh(uv))$ |
| | PMOS3 | $(e^{(u^2-v)/10} \sin v,\ \cos(uv),\ u^2 + v^2)$ |
| | PMOS4 | $(\tanh(u+v^2),\ \sin(u^2 v),\ \cos(u-v^2)\log(1+u^2))$ |
| | PMOS5 | $(u \cos(v^2),\ u \sin v,\ \sin(u^2+v))$ |
| | PMOS6 | $(\sinh(uv/5),\ \cos(u-v^2),\ e^{-u^2/10} \tanh v)$ |
| | PMOS7 | $(u \sin(v^2),\ v \cos u,\ \log(1+u^2+v^2)\sin u)$ |
| | PMOS8 | $(\tanh(u^2+v)\cos v,\ \sin(uv^2),\ u^2 e^{-v/5})$ |
| | PMOS9 | $(e^{(u-v)/5} \sin(u^2),\ \cos(v^2-u),\ u \tanh(v^2))$ |

*Table 7 - continued from previous page*

| Category | ID | Equations |
|---|---|---|
| | PMOS10 | $(\sin(\frac{u^2 v}{10}),\ v\cos u,\ \log(1+u^2)\sinh(v/5))$ |
| | PMOS11 | $(\log(1+u^2)\cos(v^2),\ \sin(u+v),\ u\,e^{v^2/10})$ |
| | PMOS12 | $(\frac{u^3 \sin v}{100},\ \cos(uv^2),\ \tanh(u-v^2))$ |
| | PMOS13 | $(\sin(\frac{u^2}{v^2+1}),\ e^{-v}\cos u,\ \sinh(v^2))$ |
| | PMOS14 | $((5+v\cos(u/2))\sin u,\ (5+v\cos(u/2))\cos u,\ \ v\sin(u/2))$ |
| | PMOS15 | $((5+\sin(uv))\cos u\sin v,\ (5+\sin(uv))\sin u\sin v,\ (5+\sin(uv))\cos v)$ |
| | PMOS16 | $(\cos u\sin v,\ \sin u\sin v,\ \cos v+\frac{u}{2})$ |
| | PMOS17 | $(u,\ v,\ \sin\sqrt{u^2+v^2}+\frac{\cos u\sin v}{5})$ |
| | PMOS18 | $(\cos u\,(5+\sin 3v),\ \sin u\,(5+\sin 3v),\ \cos(3v)+u/2)$ |
| | PMOS19 | $(\sin(2u)\cos^2 v,\cos(2u)\sin v,\sin u\cos v)$ |
| | PMOS20 | $(\sinh(u/5)\cos v,\cosh(u/5)\sin v,\tanh(v)\cos u)$ |
| | PMOS21 | $(\sin(u^2+v)\,e^{-v},\ \cos(uv)\,\log(1+|v|),\ \frac{\sin(uv^2)}{1+u^2})$ |
| | PMOS22 | $(\cosh(u+v^2),\ \sinh(uv),\ \tanh(u^2-v)\cos v)$ |
| | PMOS23 | $(e^{u-v^2}\sin u,\ u^2\cos v,\ \log(1+u^2+v^2))$ |
| | PMOS24 | $(\sin(u^2)\,v,\ \cos(v^2)\,u,\ u\,e^{-v})$ |
| | PMOS25 | $(u^3-v^2,\ \cos(uv^2),\ \tanh(u-v)\log(1+u^2))$ |
| | PMOS26 | $((u^2+v^2)\sin u, (u^2+v^2)\cos v,\ \sqrt{u^2+v^2}\,\cos\sqrt{u^2+v^2})$ |
| | PMOS27 | $(\log(1+u^2)\cos v,\ \sin(u+v^2),\ u^2\tanh v)$ |
| | PMOS28 | $(\tanh(u^2)\sin v,\ u\,e^{-v^2},\ \frac{\cos(uv^2)}{1+u^2})$ |
| | PMOS29 | $(\cos(u^2+v)\,e^{u/5},\ \sin(v^2-u),\ u\,\log(1+v^2)\tanh u)$ |
| | PMOS30 | $(u\cos(v^2),\ u\sin v,\ e^{(u-v^2)/5})$ |

