# OpenReview forum: "SURFACEBENCH: A Geometry-Aware Benchmark for Symbolic Surface Discovery"
_TMLR — Accepted by TMLR_

### Review · Reviewer_xQZ2 · 2025-12-07

**Summary Of Contributions:**

Summary of Contributions:

- New benchmark SurfaceBench for symbolic surface discovery with 183 tasks across 15 categories. Tasks include explicit, implicit, and parametric equations.
- The authors consider Chamfer and Hausdorff distances to evaluate the quality of discovered equations.
- Comprehensive comparison of LLM-based vs traditional symbolic regression methods.

Strengths:

- The benchmark is well-designed to test the capabilities of symbolic regression methods in discovering equations for 3D surfaces.
- Interesting use of geometry-aware metrics for evaluation.
- Interesting results showing the limitations of current LLM-based methods.

Weaknesses:

- Limited novelty: In essence, a new table of symbolic equations (Table 6) are proposed, and the rest of the paper is an experimental comparison of existing methods on this benchmark.

**Audience:**

Yes

**Audience Explanation:**

The community interested in symbolic regression and equation discovery would find this paper relevant. The proposed benchmark, SurfaceBench, addresses a gap in existing benchmarks by focusing on 3D surfaces and incorporating geometry-aware metrics. The comprehensive comparison of LLM-based and traditional methods provides valuable insights into the current state of the field and highlights areas for future research.

**Broader Impact Concerns:**

No concerns.

**Claims And Evidence:**

Yes

**Claims Explanation:**

The authors provide a comprehensive set of experiments comparing various LLM-based and traditional symbolic regression methods on their proposed benchmark. The results are presented clearly, with appropriate metrics (Chamfer and Hausdorff distances) to evaluate the quality of the discovered equations. These comparisons are very relevant to the symbolic regression community and provide convincing evidence for the claims made in the paper.

**Requested Changes:**

- Please provide more details on how the traditional symbolic regression methods are run for the implicit and parametric equations. This would help readers better understand the experimental setup and results.

---

> ### Author Response · Authors · 2026-02-01
> **Response to reviewer xQZ2**
>
> Thank you for the constructive feedback. Please find our responses below.
> >Limited novelty: In essence, a new table of symbolic equations (Table 6) are proposed, and the rest of the paper is an experimental comparison of existing methods on this benchmark.
>
> We respectfully disagree with the reviewer that the contribution of SurfaceBench is limited to introducing a new table of equations. Prior symbolic regression benchmarks fall largely into two categories, each with fundamental and well-known limitations. Benchmarks such as AI Feynman draw heavily from canonical equations in existing literature and textbooks, which makes them susceptible to memorization by modern LLM-based methods and limits their ability to probe compositional or geometric reasoning. More recent benchmarks such as SRBench and LLM-SRBench also try to mitigate this memorization issue by generating large numbers of synthetic equations, but these expressions lack grounding in real scientific modeling contexts. So while it may seem LLMs might be able to leverage their world knowledge, since the equations are disconnected from scientific fields, they cannot leverage their priors.
> Motivated by these limitations in recent benchmarks, SurfaceBench is designed to enable empirical insights that are not accessible in existing benchmarks. For example, we find that only two frameworks: PySR and OpenEvolve can be meaningfully evaluated on parametric surfaces, exposing a fundamental limitation of most symbolic regression methods: the lack of support for coupled multi-equation discovery. Beyond capability gaps, SurfaceBench reveals systematic failure modes through its geometry-aware evaluation and failure analysis. Our results show a consistent disconnect between symbolic accuracy and geometric correctness, where LLM-based methods often identify plausible symbolic families but fail to produce geometrically accurate surfaces. The failure taxonomy further isolates two dominant weaknesses of current LLM-based frameworks: (i) search failures, where incorrect functional families are selected, and (ii) equation fitting failures, where structurally correct expressions fail to converge to accurate parameterizations. These failure modes are largely invisible in scalar benchmarks and string-based evaluations. Together with controlled ablations on noise, OOD extrapolation, and domain priors, these findings demonstrate that SurfaceBench is not merely an expanded equation list, but a benchmark that exposes previously unmeasured limitations and design gaps in state-of-the-art symbolic regression and LLM-guided discovery systems.
>
> >Please provide more details on how the traditional symbolic regression methods are run for the implicit and parametric equations. This would help readers better understand the experimental setup and results.
>
> We thank the reviewer for bringing this to our attention. For implicit equations of the form f(x,y,z)=0, we cast surface discovery as a scalar regression problem. We uniformly sample 3D points from a fixed domain and use the ground-truth implicit function values as regression targets. Traditional symbolic regression methods are trained to recover a scalar expression f^(x,y,z) by minimizing mean squared error on these samples. Surface quality is then evaluated by extracting the zero-level set f^(x,y,z)=0 via dense sampling and comparing the resulting surface to the ground truth using Chamfer and Hausdorff distances.
> Parametric surfaces require discovering multiple coupled equations (x(u,v),y(u,v),z(u,v)) which most traditional SR methods do not natively support. We therefore evaluate only PySR and OpenEvolve on parametric tasks. OpenEvolve directly supports multi-equation settings and jointly optimizes all coordinate functions. PySR is applied by learning x^(u,v),y^(u,v), and z^(u,v) independently from identical (u,v) samples and operator sets; these equations are evaluated jointly at the surface level using geometry-based metrics. Other SR baselines are excluded from parametric evaluation due to the lack of stable multi-output support.
>
> We have included the above explanation in the appendix for clarity.

---

### Review · Reviewer_w8TU · 2025-12-11

**Summary Of Contributions:**

The authors argue that LLM-based approaches for Symbolic Regression (SR) significantly underperform compared to traditional techniques. To test this hypothesis, the paper introduces SURFACEBENCH, a large-scale benchmark designed to evaluate equation discovery for 3D surfaces. The study assesses various state-of-the-art frameworks and indicates that LLM-based methods struggle with geometric reconstruction and generalization across diverse surface representations.

### Strengths:

- The paper contains a wider variety of equations considered compared to previous Symbolic regression benchmarks. It also incorporates explicit, implicit, and parametric representations, which is a significant expansion over previous work.
- Includes various ablation studies (such as the effect of noise, OOD generalization, Priors as prompts to LLM-methods).

### Weaknesses:

- Some metric descriptions are easy to miss. Symbolic accuracy is first defined briefly in Section 3.2 as using an LLM based equivalence check that follows LLM SRBench, and more details appear later in Appendix A.1 with an example prompt. A link to Appendix A.1 in the main text would help.
- In Tables 1 and 2, it is hard to visually scan which methods perform best among LLM based methods versus non LLM baselines for each metric. Some simple visual cues, such as bolding the best method within each block, would make the tables easier to parse (similar to Table 1 in LLM-SRBench).
- The legend of Fig. 6 is poorly explained. A short explanation in the caption that maps colors or line styles to specific error categories or example equations would improve readability.

**Audience:**

Yes

**Audience Explanation:**

SurfaceBench sits at a very active intersection of topics that are central to TMLR. It combines symbolic regression, scientific machine learning, and agentic or self evolving LLMs. The benchmark is carefully documented, and the authors plan to release the dataset and evaluation pipeline, which can become a shared testbed for many future methods.

**Broader Impact Concerns:**

I do not see major negative societal impacts specific to this work. The benchmark and code will likely encourage more rigorous evaluation and improve the reliability of symbolic regression and equation discovery methods in scientific domains.

**Claims And Evidence:**

Yes

**Claims Explanation:**

- The experiments on SURFACEBENCH effectively highlight the current limitations of LLM-based symbolic regression. The data shows that these methods achieve low recovery rates compared to non-LLM baselines even when provided with domain priors.
- For LLM-based methods Chamfer and Hausdorff distances remain relatively large in many settings, which confirms that even when the functional family is roughly correct, the fitted geometry can be quite far from the ground truth.

These results support the claim that the benchmark is challenging and that current methods are far from solving it.

**Requested Changes:**

1. (Critical) Presentation: Figure 1 currently mixes different font types/embedded text between the plots and the labels. Please standardize the fonts to ensure a consistent and professional look.
2. (Critical) Presentation: Please revise the legend or the supporting text for Figure 6. The legend currently lists specific terms like "Trignometric" and "Search Exponential" and uses green and yellow rectangular regions that are not defined in the main text.
3. (Critical) Section 5.3 describes explicit and implicit domain priors in qualitative terms. For reproducibility, please spell out the exact prompt templates and the inputs given to each LLM-based method, at least for a few representative explicit and implicit surfaces. This should include how coordinate system hints, domain labels, and symmetry hints are encoded.

---

> ### Author Response · Authors · 2026-02-01
> **Response to reviewer w8TU**
>
> Thank you for the constructive feedback. We have addressed all requested presentation and reproducibility issues in the revision.
> 1. Symbolic accuracy: We have added an explicit pointer in Section 3.2 to Appendix A.1, which specifies the LLM-based symbolic accuracy.
> 2. Tables 1 and 2: We have improved readability by bolding the best method within each block.
> 3. Figure 6: We have revised the legend and caption to define all error categories and explicitly map colors and regions to failure modes, and add a brief reference in the main text.
> 4. Figure 1: We have standardized fonts across the figure.
> 5. Section 5.3: We have added representative prompt templates, detailing coordinate hints, domain bounds, and priors used for LLM-based methods.

---

### Review · Reviewer_PWKE · 2026-01-15

**Summary Of Contributions:**

The paper introduces SurfaceBench, a benchmark for symbolic regression / equation discovery where the target is a 3D surface rather than a scalar-valued function. Tasks span explicit, implicit, and parametric surface representations and multiple scientific categories. The authors emphasize that surface discovery introduces stronger challenges than typical scalar SR, including multi-output coupling, latent coordinate systems/topology, and symbolic non-uniqueness. SurfaceBench evaluates candidates in object/geometric space by sampling predicted and ground-truth surfaces into point clouds and using Chamfer/Hausdorff distances as metrics.

**Additional Comments:**

----------- After Rebuttal ---------------

I would like to thank the authors for addressing my questions and updating the paper for clarification. I have no further questions

**Audience:**

Yes

**Audience Explanation:**

The move to 3D surfaces and geometry-aware evaluation seems compelling and likely to be useful for the symbolic regression community.

**Claims And Evidence:**

Yes

**Claims Explanation:**

* In page 3, point 2 "with equation recovery rates of only 4% for LLM-based frameworks and 6% for traditional symbolic regression methods.", I did not find these numbers in the result tables. Are these numbers supported by experiments or did I miss something?
* The authors claim that "our tasks reflect surface-level structure, resist LLM memorization through novel symbolic compositions", however, as I can see from the appendix, the equations in the benchmark are **quite common**, there is no concrete evidence shown like: overlap statistics (how many equations match/near-match AI Feynman / SRBench), similarity analyses, leakage tests. These could be difficult, but without these, “resists memorization” remains plausible but not convincingly evidenced.

**Requested Changes:**

* Page 4 Line 4: For the statement “evaluation criteria that operate in geometric rather than symbolic space”. There are two non-uniqueness here, I suppose, (1) different ways of parameterising the same function (2) different functions but same or very close in the evaluation domain, which relies on the domain for the samples. I guess as a benchmark, the domain (e.g., for train/test/ood) should also be defined for the user, otherwise, people might use different settings to benchmark their models and results in unfair comparison.
* For Section 5.1 and Section 5.2, there is only description for the resulting data, it will be useful to add your interpretations after the result descriptions, i.e., how do you interpret the results?
* Section 5.2, what do you mean by “If a model is trained on inputs sampled from [-5,5] along each axis” which results are from fine-tuned models, which are from direct query?

---

> ### Author Response · Authors · 2026-02-01
> **Response to reviewer PWKE (1)**
>
> Thank you for dedicating your time and expertise to review our submission. Please find our responses below.
>
> > In page 3, point 2 "with equation recovery rates of only 4% for LLM-based frameworks and 6% for traditional symbolic regression methods.", I did not find these numbers in the result tables. Are these numbers supported by experiments or did I miss something?
>
> The reported 4% (LLM-based) and 6% (traditional SR) figures correspond to exact equation recovery, i.e., cases where the discovered equation exactly matches the ground-truth analytic form (up to trivial simplifications), aggregated across all SurfaceBench tasks for LLM and non-LLM based baselines respectively.
> These values were not included in Tables 1 and 2 because exact recovery rates are uniformly very low for all methods and therefore not informative for comparison, the tables focus instead on more discriminative metrics. Thank you for bringing this to our attention which indeed could be confusing from current writing. We have added clarification on this point in the section 4 to the updated version of manuscript.
>
>
> > The authors claim that "our tasks reflect surface-level structure, resist LLM memorization through novel symbolic compositions", however, as I can see from the appendix, the equations in the benchmark are quite common, there is no concrete evidence shown like: overlap statistics (how many equations match/near-match AI Feynman / SRBench), similarity analyses, leakage tests. These could be difficult, but without these, “resists memorization” remains plausible but not convincingly evidenced.
>
> Thank you for raising thoughtful question. We would like to clarify that this is already addressed in SurfaceBench by design, as described in Section 4 and Figure 2. Specifically, SurfaceBench does not reuse canonical equations verbatim. Instead, equations from scientific domains are used only as physically interpretable seeds, which are then systematically transformed through controlled compositional augmentations (functional nesting, term blending, coordinate reparameterization, and operator substitution). These augmentations deliberately produce non-canonical symbolic forms that preserve analytic meaning while preventing template recall. Each generated equation is subsequently subjected to novelty checks against prior benchmarks (AI Feynman, SRBench, LLMSRBench), and analytic solvability and stability checks, with redundant or trivial forms discarded.
>
> To make this concrete, we have additionally conducted experiments during the rebuttal period and report an explicit overlap analysis. For each equation, we compute two complementary metrics. Operator similarity is measured as the similarity between the sets of symbolic operators in the expressions (e.g., {+, *, sin, exp, pow}), capturing overlap in operator usage while ignoring structure. AST signature similarity is computed by extracting a normalized abstract syntax tree (AST) signature that summarizes structural properties of the expression, including operator composition, function types, tree depth, and node counts; similarity is then computed between signatures using a normalized similarity score. For each SurfaceBench equation, we report the maximum similarity score against all equations in prior benchmarks under each metric.Using these, we observe low structural overlap with existing benchmarks (AI-Feynman, SRBench, LLMSRBench): operator similarity has a mean of 0.284 (0% ≥ 0.7), and AST signature similarity has a mean of 0.407, with no equations above 0.65. No exact or near-duplicate equations are observed. These results indicate that SurfaceBench equations are structurally distinct from prior benchmarks.
>
> | Benchmark   |                                                                                                              Operator Similarity |                                                                                                              AST similarity |
> | ----- | --- | ---- |
> | AI Feynman  | 0.118 |  0.191 |
> | SRBench     |    0.138         | 0.272   |
> | LLM-SRBench |  0.284  | 0.407 |
>
> We have also added Appendix A.2

---

> ### Author Response · Authors · 2026-02-01
> **Response to reviewer PWKE (2)**
>
> >Page 4 Line 4: For the statement “evaluation criteria that operate in geometric rather than symbolic space”. There are two non-uniqueness here, I suppose, (1) different ways of parameterising the same function (2) different functions but same or very close in the evaluation domain, which relies on the domain for the samples. I guess as a benchmark, the domain (e.g., for train/test/ood) should also be defined for the user, otherwise, people might use different settings to benchmark their models and results in unfair comparison.
>
> The reviewer’s understanding is correct, and we agree with the concern. Geometry-based evaluation is meaningful only with respect to a fixed evaluation domain, and without a prescribed domain, comparisons would indeed be ambiguous or unfair. Importantly, SurfaceBench already addresses this as part of the benchmark definition.
> Specifically, the training, in-domain test, and out-of-domain (OOD) ranges are explicitly defined, and all Chamfer and Hausdorff distances are computed using samples drawn exclusively from these benchmark-specified domains. This is described in Section 5.2, where OOD evaluation is defined as a strict range shift relative to the training domain (e.g., equation fitting on [−5,5] and testing on [−10,−5]∪[5,10]). These domains are used consistently across all methods, and users are not free to choose alternative evaluation ranges.
> In response to reviewer’s question, we have revised the text in Section 2.1 point 4 to explicitly state that geometry-based evaluation in SurfaceBench is always performed on fixed, benchmark-defined ID and OOD domains, making this assumption clear and avoiding any ambiguity. Kindly note train/test here implies samples which are used for equation fitting and evaluation respectively.
>
> >For Section 5.1 and Section 5.2, there is only description for the resulting data, it will be useful to add your interpretations after the result descriptions, i.e., how do you interpret the results?
>
> Thank you for the helpful suggestion. We have added more clarification on the result interpretation in the updated version of paper.
>
>
> >Section 5.2, what do you mean by “If a model is trained on inputs sampled from [-5,5] along each axis” which results are from fine-tuned models, which are from direct query?
>
> Thank you for the thoughtful question. In the updated manuscript, we have clarified that we do not train or fine-tune any model on SurfaceBench.
> In Section 5.2, “inputs sampled from [-5,5]” only specifies the domain from which we sample (x,y) points used to run each method’s standard equation-discovery / fitting procedure (i.e., the optimization/search for equations). To clarify both the train and test samples are from the same range. Train samples refer to the data points used for fitting procedure and test samples refer to the data points used for evaluation.

---

### Review · Reviewer_vp37 · 2026-02-02

**Summary Of Contributions:**

The paper claims that existing benchmarks for LLM-based symbolic regression (SR) methods often focus on scalar algebraic equations while not capturing the multivariate and geometrically structured nature of equations common in practical scientific tasks. Therefore, it introduces a comprehensive benchmark that includes surface equations from scientific modeling domains. Existing LLM-based methods are evaluated on this new benchmark and are shown to suffer from unsatisfactory overall performance. As a result, the proposed benchmark is a meaningful contribution, highlighting the many unaddressed challenges in LLM-based symbolic regression.

**Audience:**

Yes

**Audience Explanation:**

The paper raises concerns in the field of LLM-based symbolic regression, an increasingly popular direction in the past one to two years. The proposed benchmark provides a meaningful challenge for future work in this domain to target.

**Claims And Evidence:**

Yes

**Claims Explanation:**

The experimental findings are described in detail. The proposed datasets and the evaluated methods are clearly documented. Overall, the results demonstrate the main argument that existing LLM-based SR frameworks generally cannot perform well across multiple types of equation representations and evaluation metrics.

**Requested Changes:**

**The symbolic accuracy metric.** The paper now uses an LLM-based equivalence check following LLM-SR, which is fine. However, from my own experience, the LLM evaluator is prone to errors, e.g., when the equation contains long textual parts that are not mathematically significant (1e-8 * (a-really-complicated-function), for example). I request that the authors look into some evaluation results and comment on the overall quality of the LLM symbolic evaluator, and also possibly discuss the potential alternatives for this metric, e.g., by purely symbolic calculations.

---

> ### Author Response · Authors · 2026-02-06
> **Response to Reviewer vp37**
>
> We thank the reviewer for dedicating their time and expertise to review our submission. Please find our responses below.
>
> > The symbolic accuracy metric. The paper now uses an LLM-based equivalence check following LLM-SR, which is fine. However, from my own experience, the LLM evaluator is prone to errors, e.g., when the equation contains long textual parts that are not mathematically significant (1e-8 * (a-really-complicated-function), for example). I request that the authors look into some evaluation results and comment on the overall quality of the LLM symbolic evaluator, and also possibly discuss the potential alternatives for this metric, e.g., by purely symbolic calculations.
>
> We thank the reviewer for raising this point. We inspected a subset of evaluation results to assess the quality of our evaluation metrics. We observed that the main failure mode of the LLM-based symbolic accuracy evaluator arises in cases where expressions contain long subexpressions, which can occasionally confuse the LLM; this aligns with the reviewer’s experience. Outside of such edge cases, the LLM judge was generally consistent in recognizing semantically equivalent expressions. Having said that, we found symbolic accuracy useful as a qualitative indicator of whether a method recovers a human interpretable closed form. Unlike the other metrics, which are purely fitness-based, symbolic accuracy provides signal along a different axis of equation discovery, for e.g., it helps diagnose failure cases for LLM-based methods where the method identifies the correct functional family but fails to fit parameters/coefficients accurately due to limited optimization quality.
>
> However, given the aforementioned failure mode, symbolic accuracy is intentionally not our primary correctness signal. Surface reconstruction is fundamentally a geometric task, so our main conclusions rely on geometry-aware distance metrics (e.g., Chamfer and Hausdorff), which directly measure shape fidelity and are unaffected by symbolic form artifacts. We therefore treat symbolic accuracy as a secondary diagnostic for interpretibility. This also aligns with one of our key motivations in SurfaceBench: introducing geometry-aware evaluation for equation discovery over surfaces. All of our ablations are based on either these distance-based metrics or nMSE.
>
> Regarding alternatives for symbolic accuracy, a natural option is purely symbolic equivalence checking using computer algebra systems(SymPy) or similar exact equality tests. While appealing in principle, such approaches are brittle in our setting due to the prevalence of piecewise definitions, nested transcendental functions, and parameterized expressions, which frequently lead to false negatives even when two expressions are semantically equivalent over the domain of interest. Another practical alternative is a hybrid approach that combines symbolic normalization (e.g., constant pruning and algebraic simplification) with randomized numerical equivalence checks over the input domain. However, these still lack the semantic understanding that LLMs provide. We view these directions as promising for future work. However, for the purposes of this benchmark, we adopt an LLM-based symbolic evaluator as a consistent, previously used approximation that provides a useful qualitative signal without treating as a primary correctness metric.

---

### Decision · Action_Editor_PYXT · 2026-02-12

**Recommendation:** Accept as is

**Audience:**

Yes

**Audience Explanation:**

Yes, this presents a new benchmark with distinctions from existing ones in a still active area.  All reviewers agree.

**Claims And Evidence:**

Yes

**Claims Explanation:**

Yes, the benchmarks are well documented and explained.  All reviewers agree.